# PhysVLM-AVR: Active Visual Reasoning for Multimodal Large Language Models in Physical Environments

**Weijie Zhou**[1,2,3,*]    **Xuantang Xiong**[2,*]    **Yi Peng**[3]    **Manli Tao**[3]
**Chaoyang Zhao**[3,4,†]    **Honghui Dong**[1]    **Ming Tang**[3]    **Jinqiao Wang**[3,4,†]
[1]Beijing Jiaotong University
[2]Tencent Robotics X & Futian Laboratory, Shenzhen
[3]Foundation Model Research Center, Institute of Automation, Chinese Academy of Sciences
[4]ObjectEye Inc.

## Abstract

Visual reasoning in multimodal large language models (MLLMs) has primarily been studied in static, fully observable settings, limiting their effectiveness in real-world environments where information is often incomplete due to occlusion or limited field of view. Humans, in contrast, actively explore and interact with their environment—moving, examining, and manipulating objects—to gather information through a closed-loop process integrating perception, reasoning, and action. Inspired by this human capability, we introduce the **Active Visual Reasoning (AVR)** task, extending visual reasoning to partially observable, interactive environments. AVR necessitates agents to: (1) actively acquire information via sequential physical actions, (2) integrate observations across multiple steps for coherent reasoning, and (3) dynamically adjust decisions based on evolving visual feedback. To rigorously evaluate AVR, we introduce **CLEVR-AVR**, a simulation benchmark featuring multi-round interactive environments designed to assess both reasoning correctness and information-gathering efficiency. We present **AVR-152k**, a large-scale dataset offers rich Chain-of-Thought (CoT) annotations detailing iterative reasoning for uncertainty identification, action-conditioned information gain prediction, and information-maximizing action selection, crucial for training agents in a higher-order Markov Decision Process. Building on this, we develop **PhysVLM-AVR**, an MLLM achieving state-of-the-art performance on CLEVR-AVR, embodied reasoning (OpenEQA, RoboVQA), and passive visual reasoning (GeoMath, Geometry30K). Our analysis also reveals that current embodied MLLMs, despite detecting information incompleteness, struggle to actively acquire and integrate new information through interaction, highlighting a fundamental gap in active reasoning capabilities.

## 1   Introduction

Visual reasoning, a fundamental capability in artificial intelligence, has enabled multimodal large language models (MLLMs) [1, 2, 3, 4, 5] to excel at tasks such as object counting [6, 7, 8] and visual question answering (VQA) [9, 10, 11, 12]. These abilities are crucial for developing intelligent systems that can comprehend and interact with the visual world. As MLLMs continue to evolve, their potential to understand and reason about dynamic, real-world environments is increasingly recognized, paving the way for more sophisticated and autonomous agents [13, 14, 15, 16, 17].

---

[1]Equal contribution
[2]Corresponding authors

39th Conference on Neural Information Processing Systems (NeurIPS 2025).

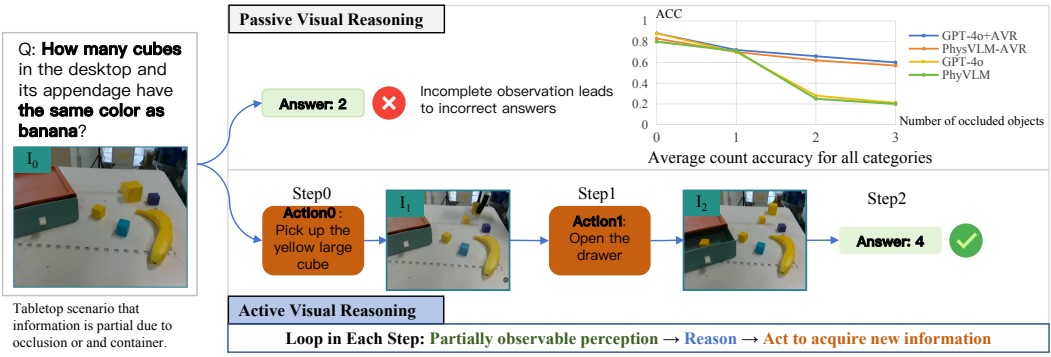

Figure 1: Passive Visual Reasoning (top) fails with partial views. Active Visual Reasoning (AVR, bottom) actively interacts to gather information for a correct answer.

Despite these advancements, current MLLM approaches to visual reasoning—including prominent tasks like VQA [18, 12, 19, 20] and spatial reasoning [21, 22, 23]—predominantly rely on static, passive visual inputs. This inherently limits their applicability in real-world physical environments where information is often partially observable due to occlusions, containment, or limited field of view [24]. As illustrated in Figure 1, a passive model observing only the initial image $I_0$ answers incorrectly, whereas an active agent interacts to uncover occluded objects across $I_1$ and $I_2$, arriving at the correct answer. And in the task of counting all object categories in the scene, the more occluded objects there are, the worse the model performs in passive visual reasoning mode. This highlights that effective reasoning in partially observable settings necessitates interaction to actively acquire missing information.

Humans, in stark contrast, naturally overcome such limitations through active perception. We instinctively move, change viewpoints, and manipulate objects to gather information needed for a specific goal. This active engagement exemplifies a core aspect of human cognition: *closing the perception-reasoning-action loop to incrementally resolve uncertainties and establish the causal links essential for complex reasoning.*

However, existing computational frameworks often suffer from paradigm fragmentation and thus fail to holistically model the complete perception-reasoning-action loop. Conventional visual reasoning tasks [25, 9, 10] (e.g., CLEVR [6]) assume complete visual inputs, requiring only a single "perception → reasoning" step without active information gathering [26, 27, 28]. While embodied question answering (EQA) tasks (e.g., OpenEQA [29], RoboVQA [30]) involve interaction, they often focus on reasoning from single, albeit potentially long, video sequences. Similarly, embodied exploration methods [31, 32, 33], also interactive, tend to prioritize task completion metrics (e.g., navigation success) over the explicit information gain needed for nuanced reasoning. The fundamental limitation across these approaches is their failure to effectively link reasoning with the strategic, sequential actions essential for information gathering. Specifically, they often overlook how an agent's actions dynamically alter available information and, crucially, how these changes should guide and refine the ongoing reasoning process, especially when multiple evidence-gathering steps are required.

Inspired by human active perception and to address these limitations, we introduce the **Active Visual Reasoning (AVR)** task. AVR extends visual reasoning to partially observable, interactive environments, bridging the gap between passive observation and active, embodied understanding. Specifically, AVR combines embodied interaction with temporal visual reasoning, requiring agents to: (1) actively gather information through sequential physical actions; (2) integrate multi-step observations for coherent reasoning; and (3) dynamically adjust decisions based on incremental visual feedback. This paradigm requires models to interpret visual data and strategically decide how to interact with the environment, thereby resolving uncertainties crucial for complex reasoning.

To establish a rigorous foundation for the AVR task and facilitate research in this domain, this paper makes the following primary contributions:

- We formally introduce and define the **Active Visual Reasoning (AVR)** task, mandating active information gathering, multi-step integration, and dynamic decision-making in partially observable settings.

- We develop **CLEVR-AVR**, a simulation benchmark featuring multi-round interactive environments. This benchmark is specifically designed to assess both the reasoning correctness and the information-gathering efficiency of agents performing AVR tasks.

- We present **AVR-152k**, a large-scale dataset with multi-level annotations designed to train agents for AVR. Its core component, AVR-Core, models the task as a higher-order Markov Decision Process and features rich Chain-of-Thought (CoT) annotations. These CoTs articulate the structured, iterative reasoning humans employ for active information seeking, including critical steps such as: (i) identifying information uncertainty, (ii) predicting action-conditioned information gain, and (iii) selecting information-maximizing actions, thereby providing explicit supervision for these decision-making processes.

- We develop and evaluate **PhysVLM-AVR**, an MLLM that achieves state-of-the-art performance on CLEVR-AVR while maintaining strong results on standard embodied visual reasoning tasks, and reveals insights into current MLLM limitations regarding active interaction.

Collectively, our work provides a foundation for developing MLLMs capable of actively reasoning about and intelligently interacting with their physical environments, thereby narrowing the gap between static visual reasoning and embodied intelligence.

## 2 Related Work

**Visual Reasoning.** Traditional visual reasoning tasks, such as Insight-V [34], CLEVR [6] and its variants (e.g., CLEVR-Math [7], MathVision [35]), have significantly advanced models' abilities in attribute recognition, counting, and spatial reasoning [36]. However, these tasks operate on static, fully observable images, lacking mechanisms for active information seeking in partially observable scenarios. Consequently, they do not model the sequential, interactive process essential for real-world understanding. AVR addresses this by requiring agents to perform sequential actions to actively acquire information necessary for reasoning.

**Embodied Question Answering.** Embodied Question Answering (EQA) tasks (e.g., OpenEQA [29], RoboVQA [30], CityEQA [37]) extend question answering to simulated embodied environments. However, by often relying on pre-captured data, these approaches limit agents to passive observation rather than active, dynamic exploration to resolve current uncertainties. As a result, the critical closed-loop interaction – where reasoning about the question drives information-seeking actions, and new information from these actions refines understanding – is frequently underemphasized. AVR, in contrast, centers on this loop, compelling agents to integrate multi-step observations acquired through their own goal-directed actions.

**Embodied Exploration.** Embodied exploration and task planning methods (e.g., Knowledge-based EQA [33], Embodied Reasoner [32], ET-Plan-Bench [38], EXPRESS-Bench [39]) enable agents to interact with their environments for broader goals like navigation or multi-step task completion. However, these approaches typically optimize for overall task success (e.g., navigation efficiency) rather than the targeted information acquisition crucial for active reasoning. Their evaluation often prioritizes general task metrics over the specific information gain pertinent to a reasoning objective. AVR, conversely, prioritizes dynamic, uncertainty-driven action selection to maximize information gain directly relevant to the goal, highlighting the link between information-seeking actions and reasoning refinement.

In summary, while existing research touches upon interaction and reasoning, AVR uniquely integrates these by emphasizing active perception driven by reasoning needs. This directly addresses robust understanding in partially observable environments, mirroring a key aspect of human cognition.

## 3 Active Visual Reasoning

### 3.1 Problem Formulation

We define **Active Visual Reasoning (AVR)** as a closed-loop paradigm where an agent must actively interact with a partially observable environment $E$ using a finite set of atomic actions $A$ to answer a question $Q$. Unlike conventional visual reasoning, AVR models reasoning as an iterative perception-reasoning-action cycle. At each timestep $t \in \{0, 1, ..., T_{\max}\}$, the agent receives a partial observation

$o_t$ and maintains an observation history $h_t = \{o_0, ..., o_t\}$. Based on $Q$ and $h_t$, the reasoning module $f_{\text{reason}}$ generates an intermediate reasoning trace:

$$\text{Think}_t = f_{\text{reason}}(Q, h_t, A). \tag{1}$$

The agent then assesses if $h_t$ is sufficient to answer $Q$. If so, it generates an answer $y_t = f_{\text{answer}}(Q, h_t)$. Otherwise, it selects an optimal information-gathering action $a_t$ to maximize expected information gain about the true answer $Y$:

$$a_t = \text{argmax}_{a_t \in A} \mathbb{E}_{o_{t+1} \sim E(h_t, a_t)} \left[ I(Y; y_{t+1} | h_{t+1}, Q) \right], \tag{2}$$

where $h_{t+1} = (h_t, o_{t+1})$. Executing $a_t$ yields $o_{t+1}$, and $h_{t+1}$ is updated. This cycle iterates until an answer is generated or $T_{\max}$ is reached.

AVR thus integrates three key components: (1) **Active Information Acquisition**: Strategically interacting to gain relevant information. (2) **Temporal Visual Integration**: Reasoning over accumulated sequential observations. (3) **Dynamic Decision-Making**: Continuously adapting beliefs and actions based on new evidence.

### 3.2 Key Challenges

AVR presents distinct challenges beyond static visual reasoning: (1) **Efficient Exploration**: Strategically exploring partially observable environments to acquire task-relevant information with minimal actions. (2) **Temporal Visual Reasoning**: Integrating and reasoning over extended observation sequences from multi-step interactions. (3) **Reasoning-Driven Actions**: Ensuring actions are guided by current understanding to purposefully reduce uncertainty, avoiding random exploration.

Addressing these multifaceted challenges necessitates dedicated benchmarks and datasets. Accordingly, the CLEVR-AVR benchmark and the AVR-152k dataset, detailed in the next section, are designed to encapsulate these difficulties. Our dataset construction, in particular, captures the sequential decision-making processes inherent in AVR, providing a structured foundation for developing and evaluating agents for these complex tasks.

## 4 CLEVR-AVR Benchmark and AVR-152k Dataset

### 4.1 CLEVR-AVR: A Benchmark for Evaluating AVR Capabilities

The **CLEVR-AVR** benchmark is a simulation-based evaluation suite designed to rigorously assess an agent's proficiency in the AVR paradigm. Leveraging the Genesis [40] physical simulation platform, it extends the classic CLEVR [6] setup into an interactive, embodied domain. This provides a controlled yet rich environment for studying the closed loop of perception, reasoning, and action fundamental to AVR. Further details on the scene types, question template structures, and the distribution of occlusion and stacking challenges within CLEVR-AVR are provided in Appendix Figures A-1, A-2, and A-3, respectively.

**Challenge of Efficient Exploration in Partial Observability.** A core tenet of AVR is overcoming incomplete information. As shown in Figure 2, CLEVR-AVR confronts agents with diverse scenarios featuring 10 occlusion types, 10 stacking types, and 10 composite scenarios combining these elements. The action space includes object manipulation and viewpoint changes, with agents required to interactively gather information to answer diverse question types. For detailed action formats and candidate generation, see Appendix Sec. A.1.

**Challenge of Temporal Visual Reasoning.** CLEVR-AVR demands multi-step integration of observations for coherent reasoning. On average, each question requires at least two reasoning steps to resolve, with some scenarios designed to necessitate 4-6 interaction steps.

**Challenge of Reasoning-Driven Actions.** Each question in CLEVR-AVR is paired with candidate responses that include both final answer options and intermediate `[Action]` options, as exemplified in Figure 2. This structure allows agents to either provide a conclusive answer or explicitly choose to interact further with the environment. Agents must assess whether current information is sufficient or if an action is necessary to reduce uncertainty, promoting purposeful, uncertainty-reducing interactions over random exploration. (Further details on action candidate generation are in Appendix Sec. A.1).

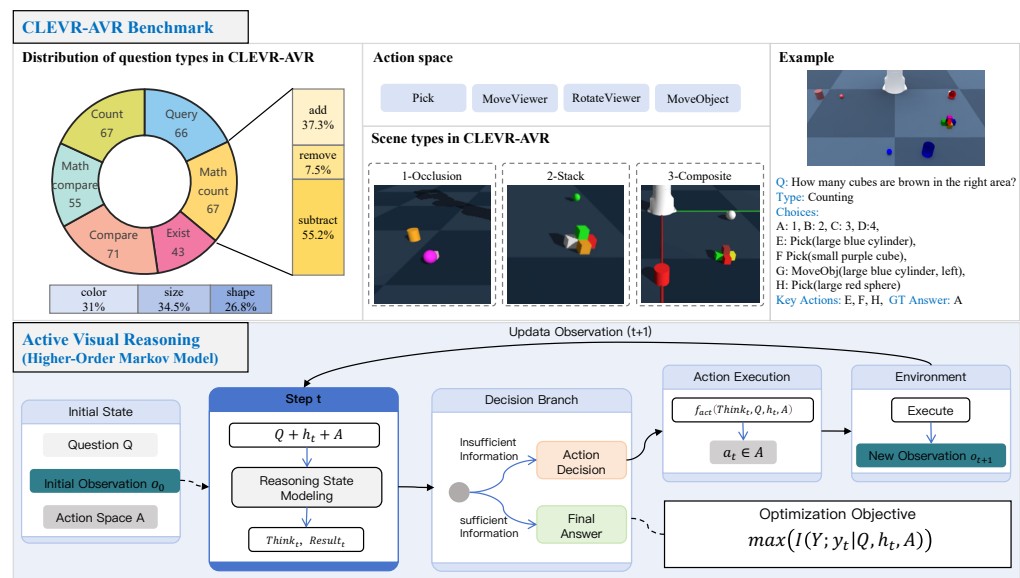

Figure 2: Top: CLEVR-AVR Simulator Benchmark, showing distributions of question types, action space, scenes, and examples. Bottom: Higher-order Markov Decision Process (MDP) paradigm for Active Visual Reasoning (AVR).

**Evaluation Metrics.**

CLEVR-AVR primarily evaluates: (1) **Information Sufficiency Judgment Accuracy** ($ACC_{ISJ}$): Accuracy in judging whether the initial observation provides sufficient information to answer the question. For example, if crucial objects for a counting task are occluded in the initial view, a correct judgment would be to acknowledge insufficiency and select an action option rather than attempting a premature final answer. (2) **Information Gain Rate** ($IGR$): This metric is derived by dividing the count of information-gaining decisions by the total number of decision steps. (3) **Final Answer Accuracy** ($ACC_{FA}$): Correctness of the final answer, with ground truth from the simulator.

An action achieves *Information Gain* if it reveals new, question-relevant visual information (e.g., uncovering hidden objects, changing viewpoints for obscured areas). This is determined by simulator ground truth: an action gains information if it unhides relevant objects/surfaces previously occluded or stacked upon. This objectively measures if the action yielded a more complete observation, crucial for tasks like counting or attribute querying needing comprehensive exploration. These metrics together comprehensively assess an agent's efficient and accurate Active Visual Reasoning (AVR).

## 4.2 AVR-152k Dataset: Modeling Active Visual Reasoning as a Higher-Order MDP

To facilitate the development of agents capable of Active Visual Reasoning (AVR)—particularly in addressing the challenges of **Efficient Exploration, Temporal Visual Reasoning, and Reasoning-Driven Actions**—we introduce the **AVR-152k dataset**. A key component of this dataset, **AVR-Core**, is specifically designed to model AVR tasks as a **higher-order Markov Decision Process (MDP)**, as illustrated in Figure 2. This MDP formulation provides a principled framework for agents to learn how to sequentially gather information and reason in interactive environments.

Within this MDP, As shown in Figure 3, AVR-Core provides rich Chain-of-Thought (CoT) annotations for the reasoning state (Think$_t$). These CoTs are meticulously structured to reflect a natural, human-like information-seeking process: (1) *Assessing Current Understanding*: evaluating the information available from observation history ($h_t$) in relation to the question ($Q$) and identifying key uncertainties or missing details. (2) *Evaluating Potential Actions*: considering available actions ($A$) and forecasting their potential to resolve identified uncertainties and yield valuable new information. (3) *Strategic Decision-Making*: based on the assessment and evaluation, deciding whether to take an information-gathering action ($a_t$) or, if uncertainty is sufficiently resolved, provide a final answer. Training with these explicit CoT annotations enables models to internalize this multi-step reasoning.

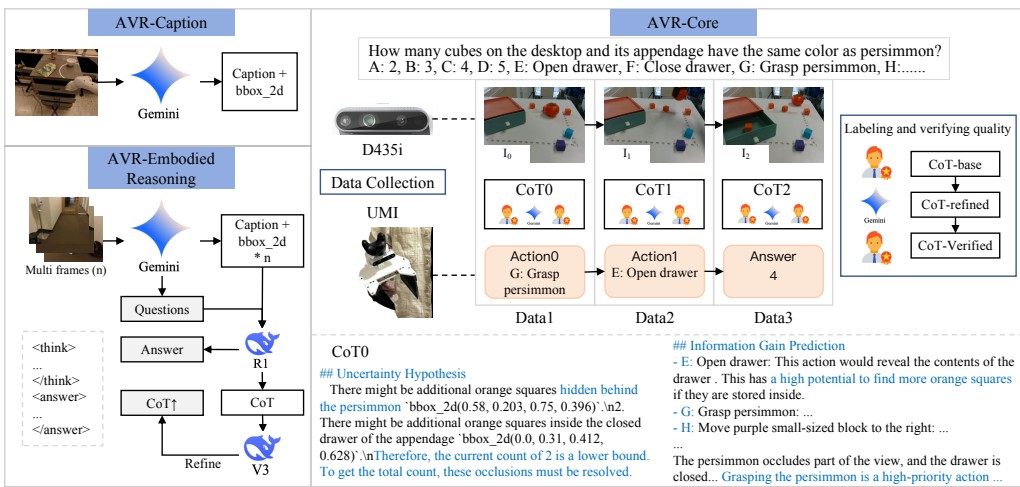

Figure 3: AVR-152k Dataset Construction. Left: Workflows for AVR-Caption and AVR-Embodied Reasoning (perception, temporal reasoning). Right: AVR-Core details including its sequential annotation, CoT supervision, and quality verification.

The full AVR-152k dataset, including its foundational subsets, supports training models to master this complex, interactive reasoning loop. As shown in Figure 3, AVR-152k comprises three subsets with progressively increasing complexity:

*AVR-Caption (100k samples):* Focuses on foundational visual perception and spatial understanding, providing dense captions with bounding boxes for indoor scenes from diverse embodied datasets (e.g., ScanNet [41], RT1 [42]). Captions were generated by Gemini-2.0-flash [43] using the prompt detailed in Appendix Figure A-4, and an example data instance is shown in Appendix Figure A-5.

*AVR-Embodied Reasoning (50k samples):* Advances to **temporal visual reasoning**. It consists of multi-image sequences paired with questions requiring spatiotemporal understanding. Dense captions were generated by Gemini-2.0-flash, while DeepSeek-R1-671B [44] produced reasoning chains (see Appendix Figure A-6 for the prompt), which were subsequently optimized by DeepSeek-V3 [45] (see Appendix Figure A-7 for the refinement prompt). An illustrative example from this subset is provided in Appendix Figure A-8.

*AVR-Core (2k samples):* Directly instantiates the higher-order MDP for AVR. Data was collected using UMI [46] devices interacting with 640 real-world tabletop settings. As shown in Figure 3, each sample in AVR-Core contains expert-structured CoT annotations for the $\text{Think}_t$ state, initially authored by human experts through a multi-step process (details in Appendix Sec. B.1), and subsequently refined by Gemini (see Appendix Figure A-9 for the refinement prompt). These annotations explicitly demonstrate the human-like reasoning process. They detail how to assess uncertainty, predict information gain from potential actions, and articulate the rationale for selecting an action or providing a final answer. This iterative process is exemplified in the multi-step interactive reasoning sequences shown in Appendix Figures A-10, A-11, and A-12. Questions typically involve several reasoning-and-action steps (avg. 3.2), requiring models to repeatedly apply this iterative decision-making process. AVR-Core underwent rigorous validation: expert annotation, logical consistency verification by Gemini [43], and human expert review of perception-reasoning-action validity.

The higher-order MDP embodied by AVR-Core (illustrated in Figure A-10, A-11, and A-12) formalizes this iterative, closed-loop process. Each sample in AVR-Core captures a single step of this interaction. It provides the current context, including the question $Q$, visual observation history $h_t$, past action history $a_{0:t-1}$, and the set of available actions $A$. Crucially, it contains the expert-annotated reasoning trace $\text{Think}_t$, which details the assessment of current understanding, evaluation of potential actions, and strategic decision-making. As an approximation of the goal in Equation 2, This reasoning culminates in a recorded outcome: either a chosen information-gathering action $a_t \in A$ or a final answer $y_t$. If an action $a_t$ is selected, the resulting new visual observation $o_{t+1}$ from the environment $E$ is also included, forming the basis for the next step. This structured data allows models to learn the

mapping from the current state and available actions to the generation of a reasoning trace Think$_t$ and the subsequent decision.

AVR-152k, with its structured CoT annotations within an MDP framework, offers a principled approach for developing models that actively seek information for reasoning in interactive environments.

# 5 PhysVLM-AVR Model

To equip a MLLM with active visual reasoning capabilities, we develop **PhysVLM-AVR**. The architecture of PhysVLM-AVR is similar to LLaVA [2], employing Qwen2.5-3B [47] as the LLM decoder and SigLIP-400M [48] as the visual encoder. A key modification for efficient multi-image reasoning is the introduction of a max pooling layer after the visual encoder's output, reducing the number of visual tokens by a factor of three. This allows for more efficient processing of multiple visual inputs. More details of the model architecture see Appendix A-14.

We employ a multi-stage, mixed-data training strategy to progressively build the generalizable active visual reasoning capabilities of PhysVLM-AVR:

- **Stage 1: Alignment.** In this initial stage, we focus on aligning visual features with the language model. We train only the connector layers (2xMLP) using the LLaVA-Pretrain [2] dataset.
- **Stage 2.1: Single-Image Understanding.** To develop foundational image comprehension, all parameters are fine-tuned using the LLaVA-OneVision-data [3].
- **Stage 2.2: Comprehensive Visual Understanding.** We then enhance the model's broader visual understanding capabilities by training on the M4-Instruct-data [3] and our *AVR-Caption*.
- **Stage 3: General Reasoning and Active Visual Reasoning.** The final stage aims to instill general reasoning abilities and specialized active visual reasoning skills. For this, we fine-tune the model on a diverse mixture of datasets: Reason-RFT-129k [25], AM-DeepSeek-R1-Distilled-100k [49], our *AVR-Embodied Reasoning* and *AVR-Core* datasets.

This multi-stage training culminates in the **PhysVLM-AVR-3B** model. The detailed training configuration for PhysVLM-AVR-3B, including hyperparameters and software environment, is presented in Appendix Section C and Figure A-13.

# 6 Experiment

## 6.1 Experimental Setup

**Tasks.** We demonstrate the effectiveness of our dataset and model through experiments across the following three categories of tasks:

- **Active Visual Reasoning.** To demonstrate that our approach enables models to acquire active visual reasoning capabilities, we first conduct comparative experiments on the *CLEVR-AVR* (Sec. 4) benchmark. The CLEVR-AVR benchmark is built on the Genesis simulation framework, ensuring that no similar images appear in the training data.
- **Embodied Reasoning and Planning.** To demonstrate that the reasoning capabilities developed with our dataset and exemplified by PhysVLM-AVR are also effective in embodied reasoning and task planning scenarios, we include comparative experiments on two embodied benchmarks: *OpenEQA* [29] and *RoboVQA* [30].
- **Visual Reasoning.** To demonstrate the generalization reasoning capability of PhysVLM-AVR in static visual reasoning tasks, we further evaluate its performance on two visual structure perception benchmarks: *GeoMath* [50] and *Geometry3K* [51].

**Baselines.** In addition to our **PhysVLM-AVR-3B** (Sec 5), we also fine-tune Qwen2.5-VL-7B on the AVR-152K dataset to obtain **AVR-Qwen2.5-VL-7B** (fine-tune details see A-13). We compare them against four categories of models: (1) `Open-source MLLMs`: Qwen2.5-VL-7B [1] and LLaVA-OV-7B [3], representing standard multimodal foundation models. (2)`Visual Reasoning MLLMs`: R1-Onevision-7B [52] and Reason-RFT-7B [25], specialized for visual reasoning tasks. (3)

`Embodied MLLMs`: Embodied-Reasoner-7B [32] and RoboBrain-7B [13], designed for embodied reasoning tasks. (4) `API-based MLLMs`: GPT-4o [53] and Gemini-2.0-flash [43].

**Evaluation Metrics.** For CLEVR-AVR, we report $ACC_{ISJ}$, $IGR$, and $ACC_{FA}$ (Sec 4.1), measuring the correctness of judging whether the initial observation is sufficient, information gain rate of action decision and accuracy of final answers. For OpenEQA, we follow the *LLM-score* [29] (GPT-4o) evaluation protocol of the original paper. For RoboVQA, we report *BLEU1-4* scores [30] as established in the original benchmark.

Table 1: CLEVR-AVR Benchmark: Our PhysVLM-AVR-3B and AVR-Qwen2.5-VL-7B vs. various MLLM categories. Best bold, second underlined.

| Method | Occlusion | | | Stack | | | Composite | | | AVG. | | |
|---|---|---|---|---|---|---|---|---|---|---|---|---|
| | $ACC_{ISJ}$ | $IGR$ | $ACC_{FA}$ | $ACC_{ISJ}$ | $IGR$ | $ACC_{FA}$ | $ACC_{ISJ}$ | $IGR$ | $ACC_{FA}$ | $ACC_{ISJ}$ | $IGR$ | $ACC_{FA}$ |
| `Open-sourceMLLMs` | | | | | | | | | | | | |
| LLaVA-OV-7B | 0 | 0 | 0 | 0 | 0 | 0 | 0 | 0 | 0 | 0 | 0 | 0 |
| Qwen2.5-VL-7B | 0 | 0 | 0 | 7.7 | 5.8 | 7.7 | 7.1 | 5.4 | 0 | 4.9 | 3.7 | 2.6 |
| `Reasoning MLLMs` | | | | | | | | | | | | |
| R1-Onevision-7B | 0 | 0 | 0 | 6.6 | 4.9 | 3.3 | 6.1 | 6.1 | 2.0 | 4.2 | 3.7 | 1.8 |
| Reason-RFT-7B | 0 | 0 | 0 | 0 | 0 | 0 | 1.5 | 1.5 | 0.0 | 0.5 | 0.5 | 0.0 |
| `Embodied MLLMs` | | | | | | | | | | | | |
| RoboBrain-7B | 4.5 | 3.8 | 0.0 | 1.6 | 0.0 | 1.6 | 4.6 | 3.1 | 3.1 | 3.6 | 2.3 | 1.6 |
| Embodied-Reasoner-7B | 21.2 | 13.6 | 1.5 | 16.4 | 8.2 | 3.3 | 23.1 | 10.8 | 0.0 | 20.2 | 10.9 | 1.6 |
| `API-based MLLMs` | | | | | | | | | | | | |
| Gemini-2.0-flash | 50.8 | 24.4 | 33.6 | 52.6 | 27.4 | 31.0 | 56.3 | 42.0 | 25.9 | 53.2 | 31.3 | 30.2 |
| GPT-4o | 85.2 | 39.4 | **53.1** | **90.5** | **46.8** | 41.4 | 89.6 | **66.3** | **42.5** | 88.4 | **50.8** | **45.7** |
| **AVR-Qwen2.5-VL-7B** | **95.5** | **43.9** | 40.9 | 88.5 | 26.2 | 36.1 | 82.9 | 40.8 | 37.4 | 89.3 | 34.7 | 38.1 |
| **PhysVLM-AVR-3B** | 90.6 | 27.4 | 42.2 | **90.5** | 22.4 | 37.9 | **90.2** | 40.0 | 39.1 | **90.5** | 29.9 | 39.7 |

## 6.2 Results on Active Visual Reasoning Tasks

Table 1 results from CLEVR-AVR offer compelling insights into active visual reasoning and highlight our AVR framework's significance. First, the results validate our premise: **existing MLLMs, trained on static data, struggle with active reasoning in interactive, partially observable environments.** Standard open-source MLLMs (LLaVA-OV-7B, Qwen2.5-VL-7B) and passive visual reasoning models (R1-Onevision-7B, Reason-RFT-7B) show near-zero performance. This illustrates passive capabilities don't translate to AVR's dynamic demands, necessitating new paradigms.

Critically, existing embodied MLLMs reveal a fundamental limitation. While models like Embodied-Reasoner-7B can detect information incompleteness (20.2% $ACC_{ISJ}$), they largely fail to act effectively for correct reasoning (only 1.6% $ACC_{FA}$). This highlights: **current embodied agents may recognize missing information but struggle to strategically acquire and integrate it.**

In contrast, our PhysVLM-AVR-3B and the fine-tuned AVR-Qwen2.5-VL-7B excel in Information Sufficiency Judgment Accuracy ($ACC_{ISJ}$) (90.5% and 89.3%), surpassing GPT-4o (88.4%). This **validates the CoT of our AVR-Core dataset to teach the identification of uncertainty and the need for interaction.** Beyond this, AVR-Qwen2.5-VL-7B achieves a robust 39.7% average final reasoning accuracy ($ACC_{FA}$), making it the best open source model, second to GPT-4o and well ahead of other baselines. The PhysVLM-AVR-3B also shows considerable improvement (39.7% $ACC_{FA}$), further showing the impact of our data set.

However, the gap between our models' high $ACC_{ISJ}$ and their final $ACC_{FA}$ (e.g., PhysVLM-AVR-3B: 90.5% $ACC_{ISJ}$ vs. 39.7% $ACC_{FA}$) highlights AVR's central challenge: **mastering optimal action selection and multi-step information integration for coherent reasoning.** While identifying the *need* to act is learned, consistently choosing the *best* action and synthesizing information over time needs more development.

In summary, CLEVR-AVR results show existing MLLMs' difficulty with AVR. They also affirm our AVR framework (AVR-152K dataset and PhysVLM-AVR model) is a significant step towards MLLMs that can intelligently explore, gather information, and reason in physical environments.

## 6.3 Results on Embodied Reasoning and Planning Tasks

As shown in Figure 4(a) and (b), our proposed models, PhysVLM-AVR-3B and AVR-Qwen2.5-VL-7B, achieve strong performance on both the OpenEQA and RoboVQA embodied reasoning benchmarks. On OpenEQA, our models consistently outperform standard multimodal models and dedicated embodied reasoning models across all sub-tasks. Notably, even when trained with only 1/20 of the RoboVQA training set (indicated by *), our models deliver BLEU scores close to or surpassing those of fully supervised baselines. These results demonstrate that active visual reasoning and the AVR-152k dataset significantly enhance embodied reasoning and planning capabilities, even under limited data conditions.

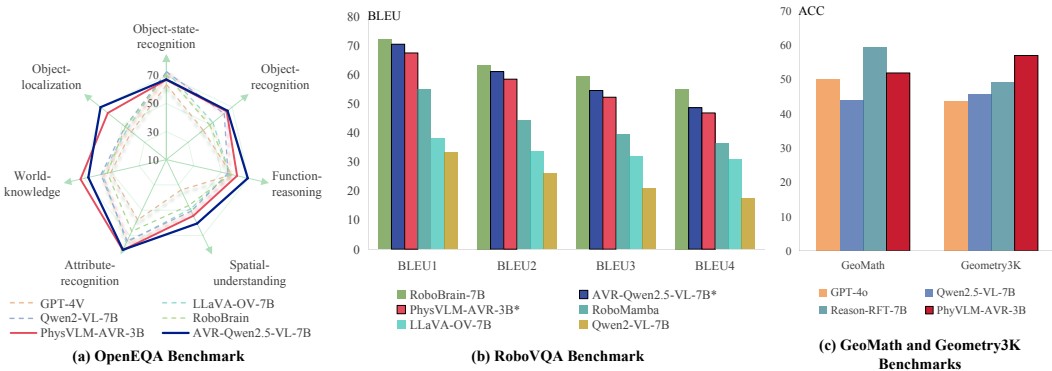

Figure 4: Results for Embodied and Visual Reasoning Tasks.

## 6.4 Results on Visual Reasoning Tasks

We further evaluate our approach on static visual reasoning benchmarks, GeoMath and Geometry3K. As shown in Figure 4(c), PhysVLM-AVR-3B achieves the highest Accuracy among all compared models, outperforming both large-scale API models (GPT-4o) and specialized visual reasoning models (Reason-RFT-7B). This indicates that our active reasoning paradigm not only benefits embodied scenarios but also generalizes well to traditional visual reasoning tasks, confirming the broad applicability and robustness of our approach.

## 6.5 Ablation Study

To assess the contributions of our AVR-Core dataset and its Chain-of-Thought (CoT) annotations, we conducted ablations on CLEVR-AVR (see Table 2). Removing the entire **AVR-Core** dataset ("w/o AVR-Core") caused a catastrophic performance collapse: $ACC_{ISJ}$ dropped from 90.5% to 16.4%, $ACC_{FA}$ from 39.1% to 2.3%, and $IGR$ from 29.9% to 11.2%. This highlights AVR-Core's fundamental role in teaching the model to actively gather information and reason iteratively within the higher-order MDP framework.

Excluding only the **Chain-of-Thought (CoT) annotations** from AVR-Core ("w/o CoT") also led to a significant decline: $ACC_{ISJ}$ fell to 47.6%, $IGR$ to 18.0%, and $ACC_{FA}$ to 16.9%. This underscores the CoTs' crucial function in providing explicit supervision for the nuanced reasoning steps of uncertainty identification, action-conditioned information gain prediction, and strategic action selection.

| Method | $ACC_{ISJ}$ | $IGR$ | $ACC_{FA}$ |
|---|---|---|---|
| Full Model | 90.5 | 29.9 | 39.7 |
| w/o CoT | 47.6 | 18.0 | 16.9 |
| w/o AVR-Core | 16.4 | 11.2 | 2.3 |

Table 2: Ablation study results.

These ablations clearly demonstrate that both the specialized AVR-Core dataset and its detailed CoT annotations are indispensable for developing effective Active Visual Reasoning capabilities.

## 7 Conclusion

In this work, we introduced Active Visual Reasoning (AVR), a novel paradigm that extends visual reasoning to interactive, partially observable environments. We developed the CLEVR-AVR bench-

mark to rigorously evaluate AVR capabilities and the AVR-152k dataset, with its core AVR-Core component providing detailed Chain-of-Thought annotations within a higher-order MDP framework, to train agents for this task. Our PhysVLM-AVR model demonstrates significant progress, achieving state-of-the-art performance on CLEVR-AVR and showing strong generalization to other embodied and static reasoning tasks. Our findings highlight that while current models can identify information incompleteness, a critical challenge remains in enabling them to strategically act to acquire and integrate new information effectively. Future work will focus on enhancing the model's ability to predict action-conditioned information gain and select optimal information-gathering actions. We also plan to explore the application of AVR to more complex real-world scenarios and investigate methods for improving sample efficiency in learning these active reasoning skills.

## Acknowledgments

This work was supported by the National Key Research and Development Program of China (No. 2022YFB4300400), the National Key R&D Program of China under Grant Nos. 2024YFE0210600 and 2022ZD0160601, and the National Natural Science Foundation of China under Grant Nos. 62176254, 62276260.

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

# A    More details of CLEVR-AVR benchmark

Figure A-1 shows examples of the three scene types in the CLEVR-AVR benchmark, Figure A-2 displays the question template settings for different question types, and Figure A-3 illustrates the distribution of occlusion and stacking quantities in each of the three scene types.

## A.1    Action Candidate Generation in CLEVR-AVR

The available action space includes Pick, Move Viewer, Rotate Viewer, and Move Object. Beyond the diverse scenarios, the benchmark incorporates a rich variety of question types, including Query, Exist, Counting, Compare, Math Counting, and Math Compare. Agents must utilize the provided Franka robotic manipulator and camera controls to actively uncover essential details, thereby tackling the challenge of efficient exploration under conditions of partial observability. The agent communicates its decision by generating text that includes its reasoning (CoT) and the selected action, formatted for example as <action>E</action>, where E would map to a specific action like Pick(yellow cube) from the candidate list.

At each interaction step in CLEVR-AVR, the agent is presented with 5-8 candidate actions. The types of actions (e.g., Pick, Move Viewer, Rotate Viewer, Move Object) are predefined. However, the target objects for actions like Pick or Move Object are dynamically selected based on the current visual scene, often including objects relevant to resolving potential occlusions or ambiguities. To rigorously test the agent's reasoning-driven action selection, approximately 3-5 of these candidate actions are intentionally designed as distractors or sub-optimal choices that would yield less information gain towards answering the question. This forces the model to not just pick any valid action, but to strategically select the one most likely to resolve its current uncertainty.

# B    More details of AVR-152k dataset

Figure A-4 shows the system prompt used for generating AVR-Caption data. An example of an AVR-Caption data instance, featuring an image and its corresponding dense caption, is presented in Figure A-5.

For the AVR-Embodied Reasoning subset, Figure A-6 displays the prompt used for DeepSeek-R1 to generate initial reasoning, and Figure A-7 illustrates the prompt for DeepSeek-V3 to refine this Chain-of-Thought (CoT) reasoning. A representative data instance from AVR-Embodied Reasoning, which includes a multi-image sequence, a question, and the associated reasoning chain, can be seen in Figure A-8.

The prompt provided to Gemini for refining human expert Chain-of-Thought annotations within the AVR-Core dataset is detailed in Figure A-9. This prompt utilizes placeholders such as {question}, {options}, {answer}, {visual_reasoning}, {hypothesis}, {gain_prediction}, and {planning}, which are filled with content from expert human annotations. To concretely illustrate the rich, multi-step interactive reasoning process captured in AVR-Core, Figures A-10, A-11, and A-12 collectively showcase a complete, sequential example from an AVR-Core instance.

## B.1    Initial Generation of Expert CoT Annotations for AVR-Core

The Chain-of-Thought (CoT) annotations in the AVR-Core dataset, prior to their refinement by large language models like Gemini, were meticulously crafted by human experts. The process involved two main stages:

**Live Task Execution and Initial Logging:** Human experts actively performed the interactive tasks using the UMI devices in real-world tabletop settings. During this live interaction, they made contemporaneous notes capturing their key reasoning steps, uncertainties identified, hypotheses about hidden information, and the rationale behind their intended actions at each decision point.

**Post-Interaction Refinement and Structuring:** After completing each task, the experts revisited their initial logs alongside the complete record of the interaction (including all visual observations and executed actions). They then elaborated on their initial notes, structuring them into the detailed, step-by-step CoT format that reflects the iterative process of assessing information, predicting gains from potential actions, and making a decision.

This human-centric initial annotation phase was crucial for ensuring that the CoTs genuinely reflected plausible and effective human reasoning strategies for active information gathering, forming a high-quality foundation for subsequent automated refinement and model training.

- Specifically, Figure A-10 depicts the initial state (Step 0) of an active visual reasoning scenario. Here, based on the initial visual observation and the posed question, the agent's CoT reflects its assessment of information insufficiency and its decision to take an action ("Move the yellow object to the left") to acquire more details.
- Figure A-11 shows the subsequent state (Step 1) after the execution of the first action. The agent re-evaluates the situation with the new visual input (IMAGE1), and its CoT again indicates the need for further information to fully resolve the question, leading to the planning of another action ("Move the green object to the right").
- Finally, Figure A-12 illustrates Step 2 of the process. After the second action and observing IMAGE2, the agent has gathered sufficient visual evidence, and its CoT culminates in providing the final answer ("Yes") to the question.

Together, these three figures (Figures A-10-A-12) highlight the core principles of AVR-Core: the iterative cycle of identifying uncertainty, predicting information gain conditioned on potential actions, and making strategic decisions, all articulated through detailed CoT annotations.

## C  More details of Model and Training

Figure A-13 shows the detailed training parameters for PhysVLM-AVR-3B and AVR-Qwen.25-VL-7B, including input image resolution, trainable parameters, batch size, maximum token length, learning rate, and number of epochs. We conducted the training on an Ubuntu server equipped with 8 * NVIDIA A800 GPUs. The main software used was PyTorch=2.6.0, transformers=3.72.0, flash attention2, and DeepSpeed. Figure A-14 showns the architecture of the PhysVLM-AVR.

## D  Baseline Model Input Prompt for CLEVR-AVR Experiments

In experiments on the CLEVR-AVR benchmark, the inputs for the compared baseline models are prompts that include AVR task instructions and cues for potential actions. The input template is as follows:

```
You are required to perform active visual reasoning:  when the
information obtained from image observations is insufficient
to answer the question, you need to make action decisions to
interact with the environment in order to acquire additional
visual information relevant to the question.  You should continue
gathering new observations until you can infer and summarize a
reliable answer based on the accumulated visual history.  Choose
either an answer to the question or an action decision option from
the options above.  Final option choice follow this format:  (your
analysis)...<answer>A/B/C/D/E/F...</answer>
```

## E  Code Availability

The code for our project is available anonymously at the following link: `https://anonymous.4open.science/r/anonymous-je99tt`.

## F  Ethical Considerations and Usage Restrictions

Large Language Models (LLMs) and related AI technologies possess the potential for significant societal impacts, both beneficial and detrimental. While they offer capabilities to enhance productivity, creativity, and access to information, it is crucial to acknowledge the inherent risks. These risks include, but are not limited to, the generation and propagation of misinformation, the amplification of

existing societal biases, potential for job displacement in certain sectors, and the possibility of misuse for malicious purposes.

The code, data, and models provided in this work are intended strictly for research and development purposes. They must not be employed in any high-risk applications or for activities that could result in harm, discrimination, infringement of rights, or any other negative societal consequences. Users of these resources bear full responsibility for ensuring their applications comply with all applicable laws, ethical guidelines, and responsible AI practices. We explicitly disclaim liability for any misuse of our code, data, or models. We encourage a cautious and ethical approach to the development and deployment of AI technologies.

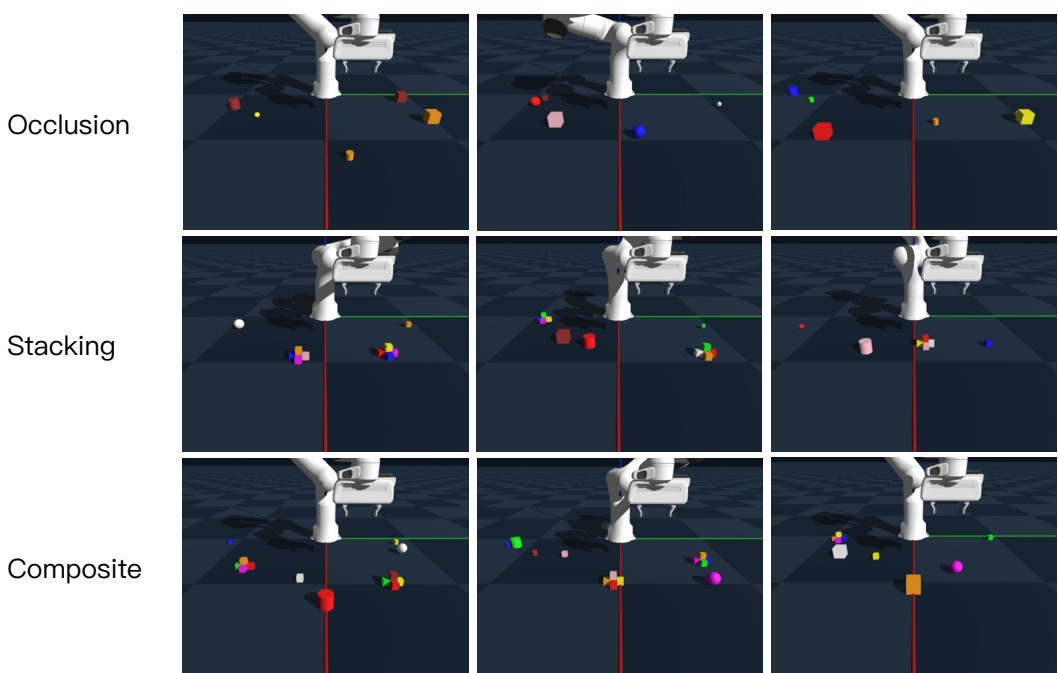

Figure A-1: Examples of the three scene types in the CLEVR-AVR benchmark.

| Question Type | Templates |
|---|---|
| Query | f"What {attr_type} is the object with the smallest center-to-center distance from the {top_obj.color} {top_obj.size} {top_obj.shape} on top?" |
| | f"What {attr_type} is the object closest to the {front_obj.color} {front_obj.size} {front_obj.shape}?" |
| | f"What is the {attr_type} of the small object on the far {direction}?" |
| Exist | f"Is there a small {getattr(back_obj, attr_type1)} that is {getattr(back_obj, attr_type2)}?" |
| Count | f"How many objects are {attr_value}?" |
| | f"In the stack with a {top_obj.color} cube on top, how many cubes are {attr_value}?" |
| | f"How many objects are {attr_value} in the {area} area?" |
| Compare | f"Is the number of {attr_value} objects in the left area and the right area equal?" |
| | f"Which area has {comparison} {attr_value} objects?" |
| | f"Is the {attr_type} of the {back_description} object the same as the {random_description} object?" |
| MathCount | f"Is the number of {attr_value} objects in the left area and the right area equal?" |
| | f"Subtract all {random_shape} {random_size} objects, how many {attr_value} objects remain?" |
| | f"Subtract all {random_shape} {random_size} objects, how many {attr_value} objects remain in the {area} area?" |
| | f"Add {random_number} {attr_value} objects, how many {attr_value} objects would there be?" |
| | f"Add {random_number} {attr_value} cubes to the stack with a {top_obj.color} cube on top, how many {attr_value} cubes would there be in the stack?" |
| | f"Subtract all {random_shape} {random_size} and {random_shape2} {random_size2} objects, how many {attr_value} objects remain?" |
| | f"Add {random_number} {attr_value} objects, how many {attr_value} objects would there be?" |
| MathCompare | f"Subtract all {random_shape} {random_size} objects, is the {attr_type} of the center cube in the stack with a {top_obj.color} cube on top the same as the {random_description} object?" |
| | f"Subtract all {random_shape} {random_size} and {random_shape2} {random_size2} objects, which area would have {comparison} {attr_value} objects?" |

Figure A-2: Question template settings for different question types in the CLEVR-AVR benchmark.

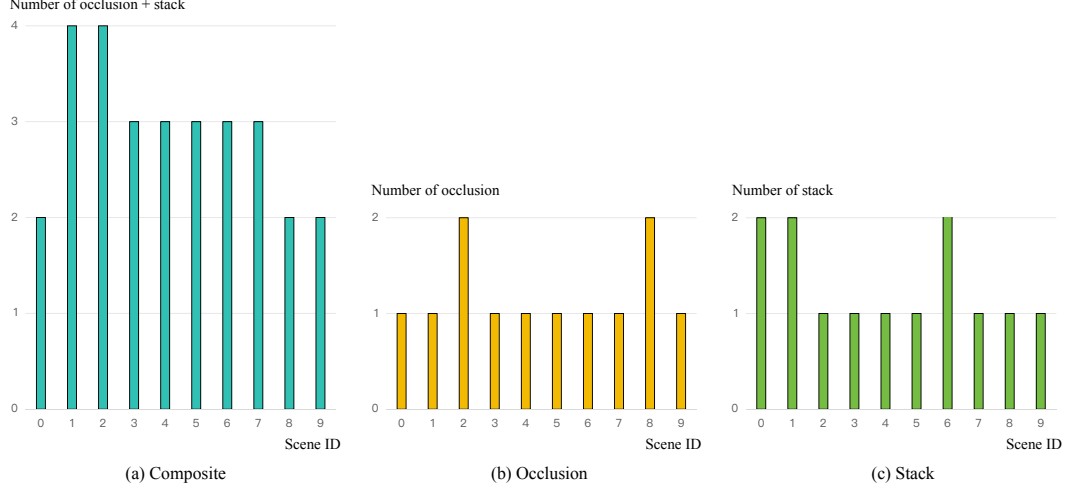

(a) Composite        (b) Occlusion        (c) Stack

Figure A-3: Distribution of occlusion and stacking quantities across the three scene types in the CLEVR-AVR benchmark.

PROMPT_CAPTION = """

Your tasks are:
### I. Provide a Detailed, Systematic, and Well-Organized Description of the Entire Scene:

Describe the scene from a macro to micro perspective, moving from the overall view to specific regions and then to individual objects in a step-by-step manner.

When describing objects, be thorough, including but not limited to:
- Color
- Shape and Structure
- Material and Texture
- State (e.g., on/off, neat/messy, in use/idle)
- Size Estimation (relative or approximate size)
- Position and Spatial Relationships (relative to other objects)
- Determine the relative hierarchical relationships between objects, their distribution in terms of distance, and layering.

**Interaction and Usage Context** (if applicable): If the combination of objects in the scene suggests a particular usage context (e.g., bowls, chopsticks, and cups on a dining table), reasonably infer and describe the possible purpose of the scene or the activity taking place.

### II. Analyze the Current Observation Perspective (Camera Position) and the Relative Positional Relationships and Spatial Distance Estimates with Other Objects, and Describe the Orientation of the Current Viewpoint:

### III. Generate 2D Object Detection Results:

- Perform 2D detection on all recognizable objects in the image.
- For each object, output its bounding box in the format `[y_min, x_min, y_max, x_max]`.
- Add a `describe` field for a detailed description of the object to distinguish between similar objects.

---
**Scene Description**
(Fill in the detailed scene description here)

**2D Detect Results**
```json
[{
    "label": "",
    "bbox": [],
    "describe": ""
 },
 ...]
```
"""

Figure A-4: System prompt for generating AVR-Caption data.

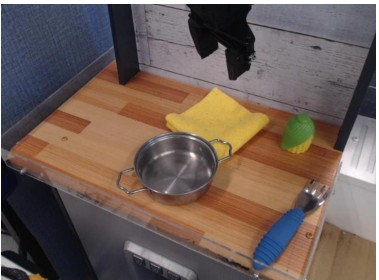

{
    "image": "avr/OXE/bridge/bridge_ac10000_0.jpg",
    "conversations": [
      {
        "from": "human",
        "value": "RGB IMAGE:\n<image>\nProvide a thorough and organized depiction of the scene utilizing RGB, starting with the overall layout and then detailing specific objects, their attributes, and spatial relationships. Incorporate any suggested interactions or contexts of use. Analyze the camera's viewpoint and the relative positions Generate 2D object detection results with normalized bounding boxes and descriptive labels in JSON format."
      },
      { "from": "gpt",
        "value": "Here's a breakdown of the scene, object detection results, and viewpoint analysis based on the provided images.\n\n**Scene Description**\n\nThe scene depicts a play kitchen setup. The backdrop consists of a wall with a faux wood paneling design on top and a blue-gray textured surface on the left. The main focus is on the countertop of the play kitchen.\n\n**Countertop:** The countertop surface has a light brown, wood-grain pattern, resembling a butcher block.  It appears to be constructed from horizontal planks. On the countertop are several play kitchen items:\n\n*  **Metal Pot:** A small, shiny silver pot sits on the left side of the countertop. It is round and has two small handles on either side. It appears to be made of stainless steel.\n*  ...... **II. Analysis of Observation Perspective**\n\nThe camera appears to be positioned at a medium height and angle, looking slightly down at the play kitchen countertop. The viewpoint is slightly to the left of center, providing a good view of all the objects on the counter. The perspective is relatively close, allowing for clear details of the objects and their arrangement.\n\n**III. 2D Detect Results**\n\n```json[\n   {\n      \"label\": \"pot\",\n      \"bbox\": [\n         0.313,\n         0.451,\n         0.611,\n         0.71\n      ],\n      \"describe\": \"Small, silver, stainless steel play pot with two handles.\"\n   },\n   {\n      \"label\": \"cloth\",\n      \"bbox\": [\n         0.435,\n         0.302,\n         0.714,\n         0.548\n      ],\n      \"describe\": \"Folded yellow cloth.\"\n   },\n   {\n      \"label\": \"corn\",\n      \"bbox\": [\n         0.742,\n         0.391,\n         0.83,\n         0.532\n      ],.... ],\n      \"describe\": \"Black plastic play faucet hanging above the countertop.\"\n   }\n]```"
      }
    ]
}

Figure A-5: Data instance of AVR-Caption.

```
PROMPT_REASONING = f"""
    Multiple images in the scene along with task-related descriptions and object bounding boxes are provided as your
visual observations. You need to combine the observations from all images to answer questions about the entire scene:
{question}

    Image descriptions: {caption}

    Note that all images are taken in the same scene, so the same object may appear in different images.
    The final answer should be a phrase, word, or other concise and clear response (if the answer cannot be found,
respond with "Unable to answer"), following the format similar to <answer>a sofa, a coffee table, ..</answer>
    """
```

Figure A-6: Prompt for DeepSeek-R1 reasoning in AVR-Embodied Reasoning data.

```
PROMPT_CoT_REFINE = f"""
    **Enhance and Modify the "Chain-of-Thought" (CoT) by Integrating Scene Captions, Adhering to the Following
Guidelines:**

    1. **STYLE TRANSFORMATION (Most Important):** Replace phrases such as "based on image description,"
"based on scene caption," "based on detection results" and "explicitly described as" with "the image shows," "I detected."
Rewrite the CoT in the FIST PERSON, focusing on image observation, analysis, and reasoning. Don't let readers feel that
you have obtained the caption and detection results of the scene with IMAGE, but rather that you have analyzed them
from the visual information of the image.

    2. **Sequential Image Summarization:** For each image (IMAGE1 to IMAGEn), provide a sequential summary of
its main content within the CoT, even if specific information related to the question isn't visible. Avoid statements like
"xxx does not appear in IMAGEn-t."

    3. **Retain Object Bounding Boxes:** Maintain the object bounding box information for each image within the
CoT to preserve spatial and contextual details. (<box>object:[xmin, ymin, xmax, ymax]</box>)

    4. **Enhance Logical Coherence:** Ensure that the CoT flows logically, with clear connections between
observations and inferences, improving the overall coherence of the reasoning process.

    5. **Maintain Tag Integrity:** Preserve the integrity of tags like `<think>`, `</think>`, `<answer>`, and
`</answer>`. Ensure that the `<answer>` tag remains concise, retaining its original brevity.

    **Scene Captions:** `{scene_captions}`

    **Chain-of-Thought (CoT):** `{cot_answer}`
    """
```

Figure A-7: Prompt for DeepSeek-V3 to refine Chain-of-Thought (CoT) reasoning in AVR-Embodied
Reasoning data.

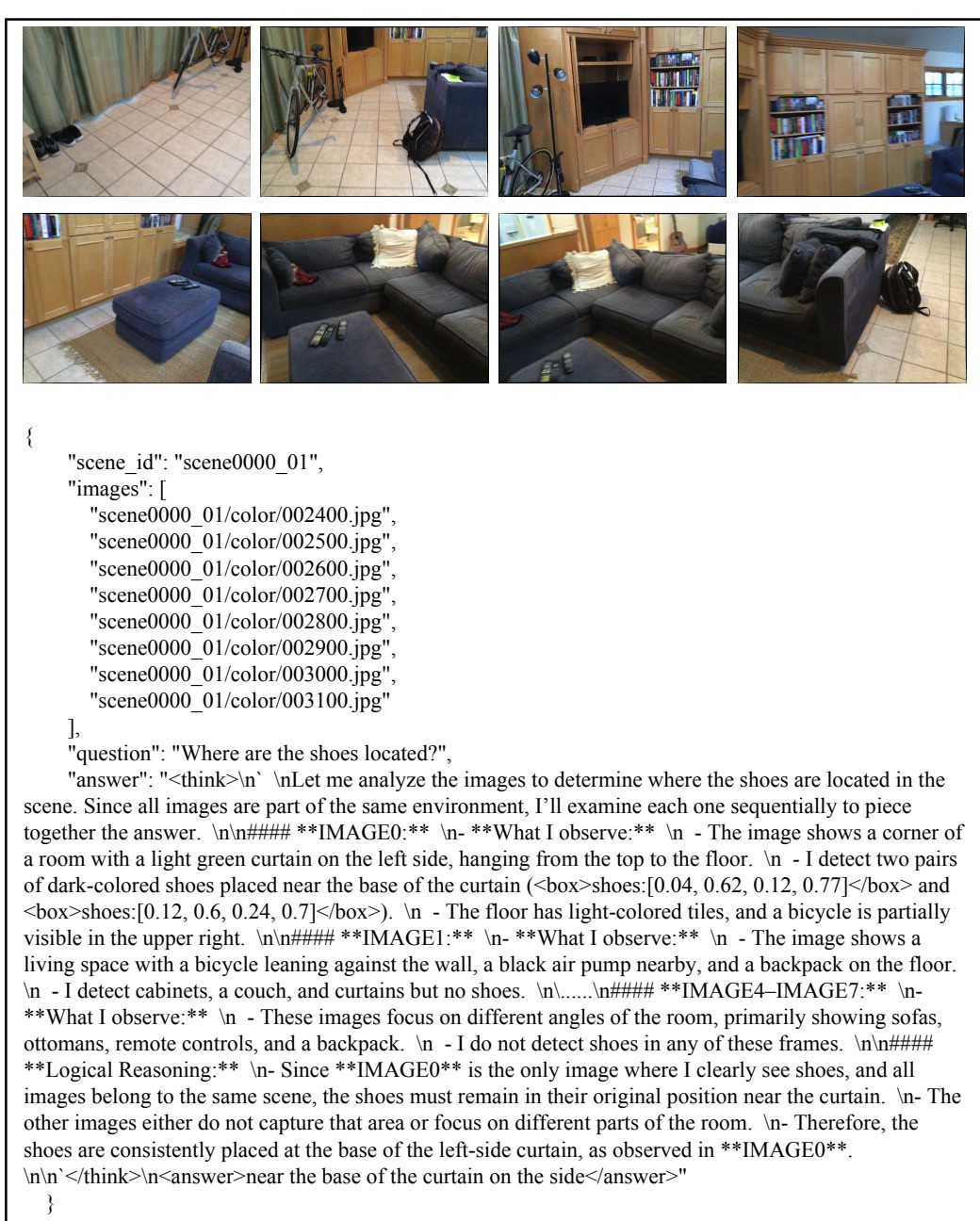

```
{
    "scene_id": "scene0000_01",
    "images": [
        "scene0000_01/color/002400.jpg",
        "scene0000_01/color/002500.jpg",
        "scene0000_01/color/002600.jpg",
        "scene0000_01/color/002700.jpg",
        "scene0000_01/color/002800.jpg",
        "scene0000_01/color/002900.jpg",
        "scene0000_01/color/003000.jpg",
        "scene0000_01/color/003100.jpg"
    ],
    "question": "Where are the shoes located?",
    "answer": "<think>\n`  \nLet me analyze the images to determine where the shoes are located in the
```
scene. Since all images are part of the same environment, I'll examine each one sequentially to piece
together the answer. \n\n#### **IMAGE0:** \n- **What I observe:** \n - The image shows a corner of
a room with a light green curtain on the left side, hanging from the top to the floor. \n - I detect two pairs
of dark-colored shoes placed near the base of the curtain (<box>shoes:[0.04, 0.62, 0.12, 0.77]</box> and
<box>shoes:[0.12, 0.6, 0.24, 0.7]</box>). \n - The floor has light-colored tiles, and a bicycle is partially
visible in the upper right. \n\n#### **IMAGE1:** \n- **What I observe:** \n - The image shows a
living space with a bicycle leaning against the wall, a black air pump nearby, and a backpack on the floor.
\n - I detect cabinets, a couch, and curtains but no shoes. \n\......\n#### **IMAGE4–IMAGE7:** \n-
**What I observe:** \n - These images focus on different angles of the room, primarily showing sofas,
ottomans, remote controls, and a backpack. \n - I do not detect shoes in any of these frames. \n\n####
**Logical Reasoning:** \n- Since **IMAGE0** is the only image where I clearly see shoes, and all
images belong to the same scene, the shoes must remain in their original position near the curtain. \n- The
other images either do not capture that area or focus on different parts of the room. \n- Therefore, the
shoes are consistently placed at the base of the left-side curtain, as observed in **IMAGE0**.
\n\n`</think>\n<answer>near the base of the curtain on the side</answer>"
```
    }
```

Figure A-8: Data instance of AVR-Embodied Reasoning.

PROMPT_CoT_REFINE = """
Active visual reasoning: When the information obtained from image observations is not enough to answer the question, action decisions need to be made to interact with the environment to obtain more visual information related to the question. New observations are continuously collected until a reliable answer can be inferred and summarized based on the accumulated visual history.

# Question: {question}
Choice: {options}
GT answer: {answer}

---

# CoT
## Goal Analysis

## Visual Reasoning
(Detailed description of items on the table, especially task-relevant items)
{visual_reasoning}

## Uncertainty Hypothesis
{hypothesis}

## Information Gain Prediction
(Analyze the information gain that would result from each action option)
{gain_prediction}

## Planning Pseudocode
(Based on the reasoning in this step, plan the subsequent logic after this action)
{planning}

## Conclusion
(Summarize the information from visual reasoning, uncertainty assessment, and information gain predictions, including key items' point and bbox_2d spatial position information.)
--

You need to enrich the CoT reasoning process. The CoT should not include GT prompts, should not change the first (# CoT) and second level (## ) structures, and should optimize the reasoning logic in each section (more efficient, rigorous). Replace the planning section with concise pseudocode. If no further action is needed, simply write END.
Use key points `point(y, x)` and 2d bounding box `bbox_2d(ymin, xmin, ymax, xmax)` in the CoT to assist visual reasoning.
"""

Figure A-9: Prompt for Gemini to refine Chain-of-Thought annotations in AVR-Core data, where {question}, {options}, {answer}, {visual_reasoning}, {hypothesis}, {gain_prediction}, and {planning} are all derived from expert human annotations.

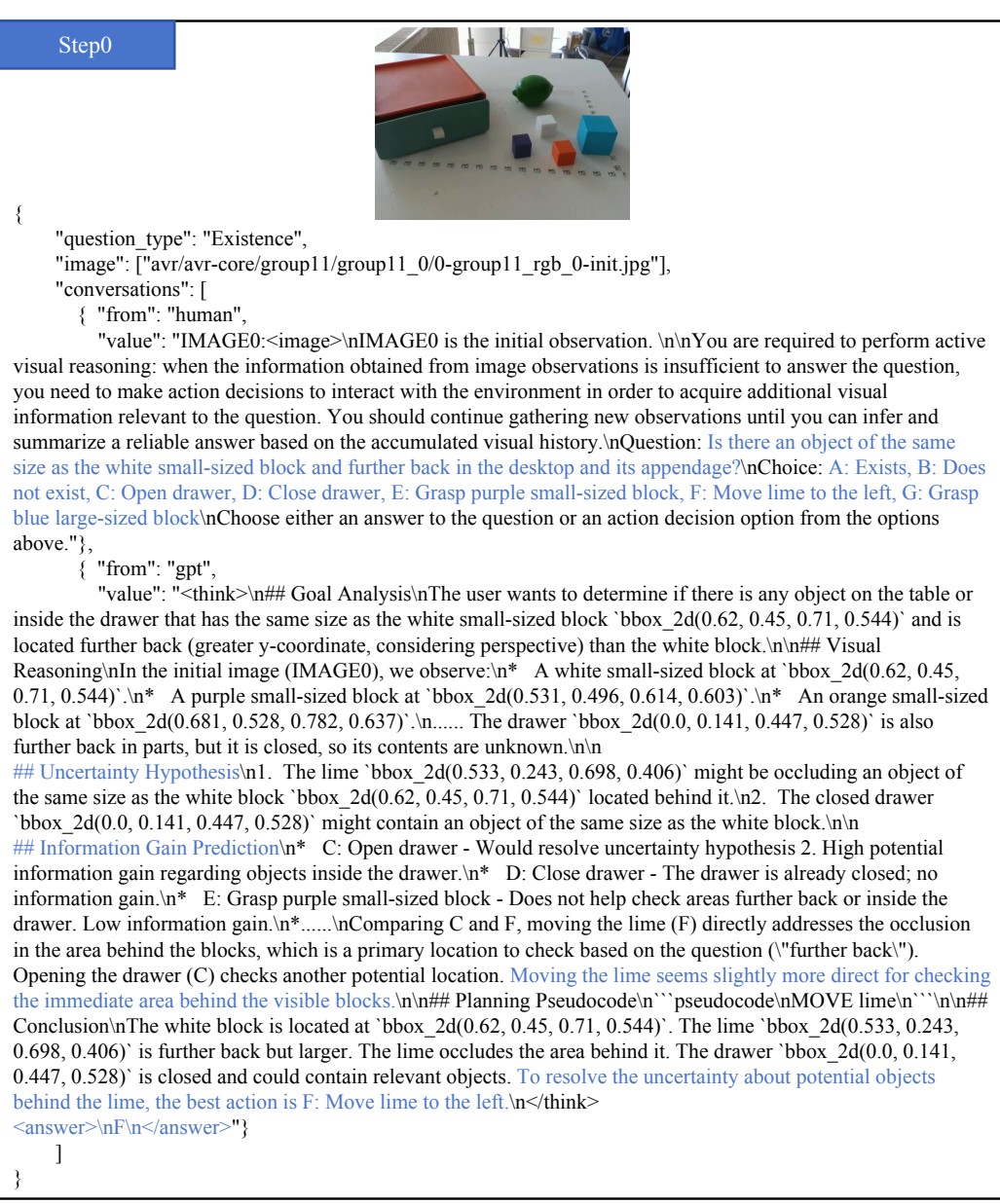

Step0

```
{
    "question_type": "Existence",
    "image": ["avr/avr-core/group11/group11_0/0-group11_rgb_0-init.jpg"],
    "conversations": [
        { "from": "human",
            "value": "IMAGE0:<image>\nIMAGE0 is the initial observation. \n\nYou are required to perform active
visual reasoning: when the information obtained from image observations is insufficient to answer the question,
you need to make action decisions to interact with the environment in order to acquire additional visual
information relevant to the question. You should continue gathering new observations until you can infer and
summarize a reliable answer based on the accumulated visual history.\nQuestion: Is there an object of the same
size as the white small-sized block and further back in the desktop and its appendage?\nChoice: A: Exists, B: Does
not exist, C: Open drawer, D: Close drawer, E: Grasp purple small-sized block, F: Move lime to the left, G: Grasp
blue large-sized block\nChoose either an answer to the question or an action decision option from the options
above."},
        { "from": "gpt",
            "value": "<think>\n## Goal Analysis\nThe user wants to determine if there is any object on the table or
inside the drawer that has the same size as the white small-sized block `bbox_2d(0.62, 0.45, 0.71, 0.544)` and is
located further back (greater y-coordinate, considering perspective) than the white block.\n\n## Visual
Reasoning\nIn the initial image (IMAGE0), we observe:\n*   A white small-sized block at `bbox_2d(0.62, 0.45,
0.71, 0.544)`.\n*   A purple small-sized block at `bbox_2d(0.531, 0.496, 0.614, 0.603)`.\n*   An orange small-sized
block at `bbox_2d(0.681, 0.528, 0.782, 0.637)`.\n...... The drawer `bbox_2d(0.0, 0.141, 0.447, 0.528)` is also
further back in parts, but it is closed, so its contents are unknown.\n\n
## Uncertainty Hypothesis\n1.   The lime `bbox_2d(0.533, 0.243, 0.698, 0.406)` might be occluding an object of
the same size as the white block `bbox_2d(0.62, 0.45, 0.71, 0.544)` located behind it.\n2.   The closed drawer
`bbox_2d(0.0, 0.141, 0.447, 0.528)` might contain an object of the same size as the white block.\n\n
## Information Gain Prediction\n*   C: Open drawer - Would resolve uncertainty hypothesis 2. High potential
information gain regarding objects inside the drawer.\n*   D: Close drawer - The drawer is already closed; no
information gain.\n*   E: Grasp purple small-sized block - Does not help check areas further back or inside the
drawer. Low information gain.\n*......\nComparing C and F, moving the lime (F) directly addresses the occlusion
in the area behind the blocks, which is a primary location to check based on the question (\"further back\").
Opening the drawer (C) checks another potential location. Moving the lime seems slightly more direct for checking
the immediate area behind the visible blocks.\n\n## Planning Pseudocode\n```pseudocode\nMOVE lime\n```\n\n##
Conclusion\nThe white block is located at `bbox_2d(0.62, 0.45, 0.71, 0.544)`. The lime `bbox_2d(0.533, 0.243,
0.698, 0.406)` is further back but larger. The lime occludes the area behind it. The drawer `bbox_2d(0.0, 0.141,
0.447, 0.528)` is closed and could contain relevant objects. To resolve the uncertainty about potential objects
behind the lime, the best action is F: Move lime to the left.\n</think>
<answer>\nF\n</answer>"}
    ]
}
```

Figure A-10: Data instance of AVR-Embodied Reasoning. Step 0 of the active visual reasoning
process, which make a action decision to get more information.

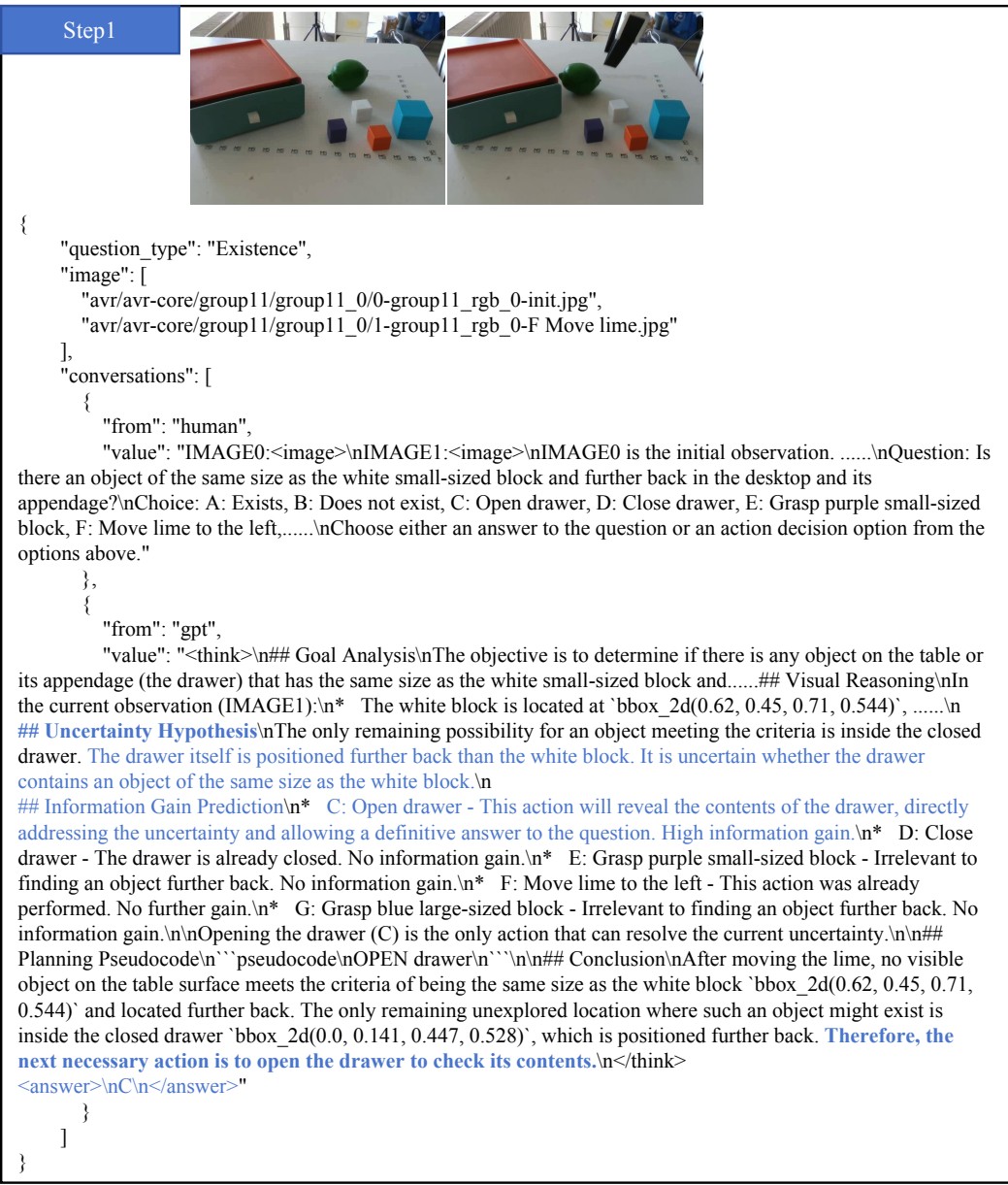

Step1

```
{
    "question_type": "Existence",
    "image": [
        "avr/avr-core/group11/group11_0/0-group11_rgb_0-init.jpg",
        "avr/avr-core/group11/group11_0/1-group11_rgb_0-F Move lime.jpg"
    ],
    "conversations": [
        {
            "from": "human",
            "value": "IMAGE0:<image>\nIMAGE1:<image>\nIMAGE0 is the initial observation. ......\nQuestion: Is
there an object of the same size as the white small-sized block and further back in the desktop and its
appendage?\nChoice: A: Exists, B: Does not exist, C: Open drawer, D: Close drawer, E: Grasp purple small-sized
block, F: Move lime to the left,......\nChoose either an answer to the question or an action decision option from the
options above."
        },
        {
            "from": "gpt",
            "value": "<think>\n## Goal Analysis\nThe objective is to determine if there is any object on the table or
its appendage (the drawer) that has the same size as the white small-sized block and......## Visual Reasoning\nIn
the current observation (IMAGE1):\n*   The white block is located at `bbox_2d(0.62, 0.45, 0.71, 0.544)`, ......\n
## Uncertainty Hypothesis\nThe only remaining possibility for an object meeting the criteria is inside the closed
drawer. The drawer itself is positioned further back than the white block. It is uncertain whether the drawer
contains an object of the same size as the white block.\n
## Information Gain Prediction\n*   C: Open drawer - This action will reveal the contents of the drawer, directly
addressing the uncertainty and allowing a definitive answer to the question. High information gain.\n*   D: Close
drawer - The drawer is already closed. No information gain.\n*   E: Grasp purple small-sized block - Irrelevant to
finding an object further back. No information gain.\n*   F: Move lime to the left - This action was already
performed. No further gain.\n*   G: Grasp blue large-sized block - Irrelevant to finding an object further back. No
information gain.\n\nOpening the drawer (C) is the only action that can resolve the current uncertainty.\n\n##
Planning Pseudocode\n```pseudocode\nOPEN drawer\n```\n\n## Conclusion\nAfter moving the lime, no visible
object on the table surface meets the criteria of being the same size as the white block `bbox_2d(0.62, 0.45, 0.71,
0.544)` and located further back. The only remaining unexplored location where such an object might exist is
inside the closed drawer `bbox_2d(0.0, 0.141, 0.447, 0.528)`, which is positioned further back. Therefore, the
next necessary action is to open the drawer to check its contents.\n</think>
<answer>\nC\n</answer>"
        }
    ]
}
```

Figure A-11: Data instance of AVR-Embodied Reasoning. Step 1 of the active visual reasoning
process, which make a action decision to get more information.

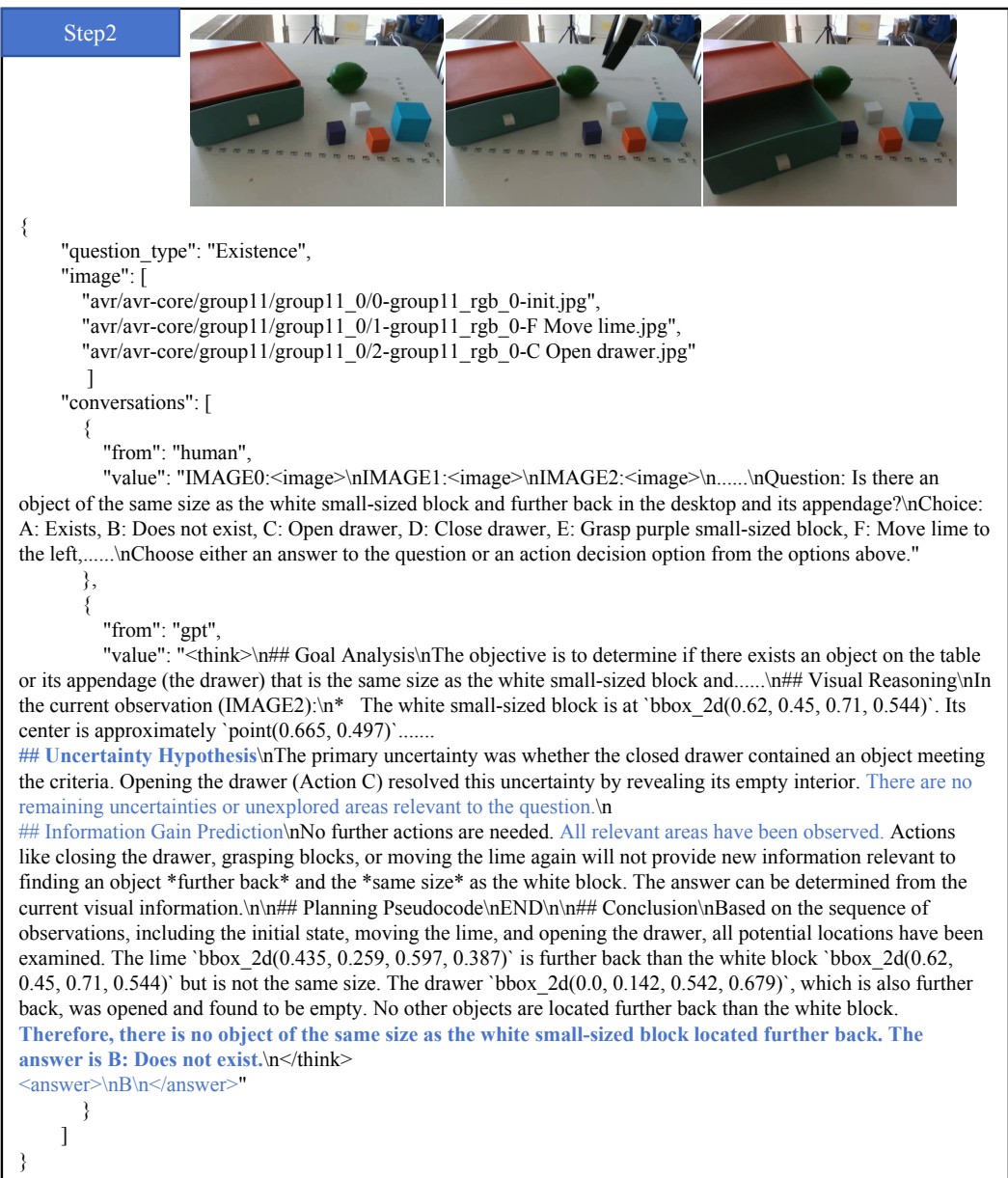

```
{
    "question_type": "Existence",
    "image": [
        "avr/avr-core/group11/group11_0/0-group11_rgb_0-init.jpg",
        "avr/avr-core/group11/group11_0/1-group11_rgb_0-F Move lime.jpg",
        "avr/avr-core/group11/group11_0/2-group11_rgb_0-C Open drawer.jpg"
    ]
    "conversations": [
        {
            "from": "human",
            "value": "IMAGE0:<image>\nIMAGE1:<image>\nIMAGE2:<image>\n......\nQuestion: Is there an
object of the same size as the white small-sized block and further back in the desktop and its appendage?\nChoice:
A: Exists, B: Does not exist, C: Open drawer, D: Close drawer, E: Grasp purple small-sized block, F: Move lime to
the left,......\nChoose either an answer to the question or an action decision option from the options above."
        },
        {
            "from": "gpt",
            "value": "<think>\n## Goal Analysis\nThe objective is to determine if there exists an object on the table
or its appendage (the drawer) that is the same size as the white small-sized block and......\n## Visual Reasoning\nIn
the current observation (IMAGE2):\n*   The white small-sized block is at `bbox_2d(0.62, 0.45, 0.71, 0.544)`. Its
center is approximately `point(0.665, 0.497)`.......
## Uncertainty Hypothesis\nThe primary uncertainty was whether the closed drawer contained an object meeting
the criteria. Opening the drawer (Action C) resolved this uncertainty by revealing its empty interior. There are no
remaining uncertainties or unexplored areas relevant to the question.\n
## Information Gain Prediction\nNo further actions are needed. All relevant areas have been observed. Actions
like closing the drawer, grasping blocks, or moving the lime again will not provide new information relevant to
finding an object *further back* and the *same size* as the white block. The answer can be determined from the
current visual information.\n\n## Planning Pseudocode\nEND\n\n## Conclusion\nBased on the sequence of
observations, including the initial state, moving the lime, and opening the drawer, all potential locations have been
examined. The lime `bbox_2d(0.435, 0.259, 0.597, 0.387)` is further back than the white block `bbox_2d(0.62,
0.45, 0.71, 0.544)` but is not the same size. The drawer `bbox_2d(0.0, 0.142, 0.542, 0.679)`, which is also further
back, was opened and found to be empty. No other objects are located further back than the white block.
Therefore, there is no object of the same size as the white small-sized block located further back. The
answer is B: Does not exist.\n</think>
<answer>\nB\n</answer>"
        }
    ]
}
```

Figure A-12: Data instance of AVR-Embodied Reasoning. Step 2 of the active visual reasoning
process, which getting the final answer.

**PhysVLM-AVR-3B training details**

Stage1: Feature Alignment

Stage2: Visual Understanding + Temporal Visual Understanding

Stage3: Basic Reasoning + Active Visual Reasoning Abilities

| | Stage1 | Stage2.1 | Stage2.2 | Stage3 |
|---|---|---|---|---|
| Image Resolution | 384 | 384 | 384 | 384 |
| Trainable | Projector | Full Model | Full Model | Full Model |
| Batch Size | 64 | 64 | 32 | 16 |
| Max length | 2048 | 2048 | 4096 | 8192 |
| LR (all parameters) | 1e-3 | 1e-5 | 1e-5 | 2e-6 |
| Epoch | 1 | 1 | 1 | 1 |

**AVR-Qwen2.5-VL-7B fine-tuning details**

| Val dataset | Datasets | Trainable | Batch Size | Max length | LR | Epoch |
|---|---|---|---|---|---|---|
| CLEVR-AVR | AVR-Core | LLM | 16 | 8192 | 5.0e-6 | 1 |
| OpenEQA and RoboVQA | All AVR-152K | LLM | 16 | 8192 | 5.0e-6 | 1 |

Figure A-13: Training configuration details for PhysVLM-AVR-3B and AVR-Qwen2.5-VL-7B.

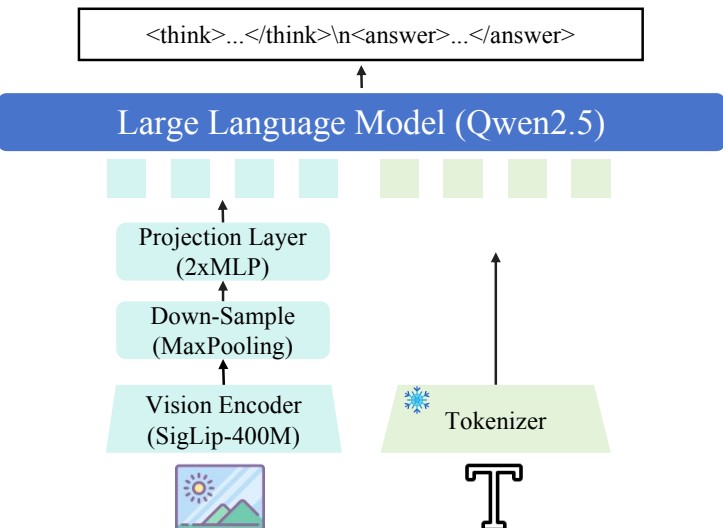

Figure A-14: Model architecture of PhysVLM-AVR.

