# OpenReview forum: "PhysVLM-AVR: Active Visual Reasoning for Multimodal Large Language Models in Physical Environments"
_NeurIPS.cc/2025/Conference — NeurIPS 2025 poster_

### Official Review · Reviewer_wcc4 · 2025-07-03

**Clarity:** 3
**Significance:** 3
**Originality:** 3
**Rating:** 4
**Confidence:** 3

**Summary:**

This paper introduces the Active Visual Reasoning (AVR) task, which evaluates visual reasoning in partially observable and interactive environments. To support this task, the authors develop the CLEVER-AVR benchmark, which provides interactive simulation environments, along with a large-scale chain-of-thought dataset, AVR-152k. Additionally, the paper presents PhysVLM-AVR, a model trained via a multi-stage paradigm that achieves SOTA performance on the proposed AVR task.

**Questions:**

- How does the training time of PhysVLM-AVR compare to that of standard fine-tuning?

**Ethical Concerns:**

["NO or VERY MINOR ethics concerns only"]

**Final Justification:**

This work contributes a benchmark beneficial to the research community. Also, the provided implementation details ensure reproducibility. Including evaluation on larger models during rebuttal further improves the quality.

**Limitations:**

Yes

**Quality:**

3

**Strengths And Weaknesses:**

### Strengths
- Active visual reasoning is an interesting and practical task with real-world use. Developing a benchmark contributes valuable resources to the research community.
- The experimental results and corresponding analysis are comprehensive.
- The paper provides detailed prompts and examples, offering clear illustrations and ensuring reproducibility.
### Weaknesses
- In Table 1, the comparison is between Qwen2.5-VL-7B (trained on AVR data) and the proposed PhysVLM-AVR-3B. It would be more informative to also include results for Qwen2.5-VL-3B trained on the same data, allowing for a fair comparison at the same model scale.
- According to Table 1, API-based models significantly outperform open-source 7B models, which score below 10% or even 0% accuracy on many tasks. It would be helpful to evaluate larger open-source models to better understand their potential on AVR.

---

> ### Author Rebuttal · Authors · 2025-07-31
>
> ### **Response to Reviewer wcc4**
>
> **Dear Reviewer wcc4,**
>
> Thank you for your valuable feedback and your suggestions for improving our experimental evaluation. Your points regarding fair model comparisons at the same scale, evaluating larger open-source models, and clarifying training time are all well-taken.
>
> We have conducted additional experiments to address your first two points and provide a detailed breakdown of the training time as requested. The results and analysis are included in our detailed responses below.
>
> Thank you for helping us strengthen our paper.
>
> ---
>
> ### **Q1: On Fair Comparison at the Same Model Scale**
>
> > **Reviewer's Comment:** In Table 1, the comparison is between Qwen2.5-VL-7B (trained on AVR data) and the proposed PhysVLM-AVR-3B. It would be more informative to also include results for Qwen2.5-VL-3B trained on the same data, allowing for a fair comparison at the same model scale.
>
> **Our Response:**
>
> Thank you for this excellent suggestion. To ensure a fair comparison, we have conducted a **new experiment by training Qwen2.5-VL-3B on our AVR data**, using the same fine-tuning protocol as the 7B version. This allows for a direct, same-scale comparison with our PhysVLM-AVR-3B.
>
> The results are as follows:
>
> | Model | Final Answer Acc. (`ACC_FA`) | Information Gain Rate (`IGR`) |
> | :--- | :---: | :---: |
> | **PhysVLM-AVR-3B (Ours)** | **39.7%** | **29.9%** |
> | AVR-Qwen2.5-VL-3B (New Baseline) | 29.6% | 28.7% |
>
> The performance gap between the two 3B models highlights two key advantages of our approach:
>
> 1.  **Architectural Optimization:** The max pooling layer in PhysVLM-AVR is specifically designed to enhance the efficiency of integrating features from multiple images, a crucial capability for multi-step reasoning that is not a native focus of the standard Qwen2.5-VL architecture.
> 2.  **Targeted Training Strategy:** Our multi-stage training curriculum places a strong emphasis on the `AVR-Embodied Reasoning` subset, which is rich in multi-image reasoning tasks. This targeted training better equips our model for the temporal reasoning demands of the AVR paradigm.
>
> ---
>
> ### **Q2: On Evaluating Larger Open-Source Models**
>
> > **Reviewer's Comment:** According to Table 1, API-based models significantly outperform open-source 7B models, which score below 10% or even 0% accuracy on many tasks. It would be helpful to evaluate larger open-source models to better understand their potential on AVR.
>
> **Our Response:**
>
> Thank you for suggesting this. To better understand the potential of larger open-source models on our task, we have conducted **a new experiment with the Qwen2.5-VL-72B model**.
>
> The results are presented below:
>
> | Model | `ACC_ISJ` | `IGR` | `ACC_FA` |
> | :--- | :---: | :---: | :---: |
> | Qwen2.5-VL-7B | 4.9% | 3.7% | 2.6% |
> | **Qwen2.5-VL-72B** | **36.0%** | **15.6%** | **16.6%** |
>
> These results clearly show that scaling up the open-source model from 7B to 72B parameters leads to a substantial improvement in performance on the AVR task. This indicates that model scale is indeed a significant factor for acquiring active visual reasoning capabilities.
>
> While there is still a gap compared to API-based models like GPT-4o, this experiment suggests that combining larger open-source backbones with our specialized multi-stage AVR training strategy could be a promising direction for future work to further close this gap.
>
> ---
>
> ### **Q3: On the Training Time Comparison**
>
> > **Reviewer's Comment:** How does the training time of PhysVLM-AVR compare to that of standard fine-tuning?
>
> **Our Response:**
>
> Thank you for your question about training efficiency. Here is a comparison of the training times:
>
> *   **PhysVLM-AVR-3B (Full Training):** The complete multi-stage training process for our model takes approximately **76 hours** on a server with 8x A800 GPUs. This includes all stages detailed in Section 4 and Appendix C of our paper, starting from feature alignment.
>
> *   **Standard Fine-tuning (AVR-Qwen2.5-VL-7B):** Fine-tuning the pre-trained Qwen2.5-VL-7B model *only* on our `AVR-Core` dataset takes approximately **2 hours** on the same hardware.
>
> It is important to note that the Qwen2.5-VL model has already undergone an extensive, multi-stage pre-training process by its developers. Our PhysVLM-AVR training starts from a much earlier stage (aligning visual and language features), which accounts for the significant difference in training time.
>
> While our multi-stage approach is more time-intensive, it improves data efficiency by allowing the model to "focus" on learning specific capabilities at each stage, leading to superior performance across multiple benchmarks. We will explore methods to improve training efficiency in our future work.

---

> > ### Comment · Reviewer_wcc4 · 2025-08-04
> >
> > Thanks for your clarifications. Most of my questions have been resolved. I will continue to follow the discussions and feedback from other reviewers. Currently, I maintain my positive score.

---

### Official Review · Reviewer_dTPo · 2025-07-03

**Clarity:** 4
**Significance:** 3
**Originality:** 3
**Rating:** 5
**Confidence:** 4

**Summary:**

The paper studies the active visual reasoning (AVR) problem with multimodal large language models (MLLMs). AVR goes beyond static visual understanding and requires actively acquiring environment information and feedback for task solving. To assess the capability of MLLMs for AVR, the authors formulate an AVR task, construct CLEVR-AVR, an AVR benchmark, introduce a dataset, AVR-152k, for developing the AVR ability of MLLMs, and train an MLLM, PhysVLM-AVR, on AVR-152k. The experimental results show that the model achieves the best performance on CLEVR-AVR while maintaining good performance on normal embodied reasoning and planning tasks, as well as static visual reasoning tasks.

**Questions:**

1. The multi-choice setup mixes candidate answers and actions; I wonder if this would introduce a bias towards choosing an answer or an action. Can the authors provide an analysis on this?
2. The comparison between AVR-QWen2.5-VL-7B and PhysVLM-AVR-3B is unfair: 1) AVR-QWen2.5-VL-7B does not rely on the more complex multi-stage training that is applied to PhysVLM-AVR-3B; 2) the model sizes are not on the same scale, i.e., 7B is more than twice as large as 3B. Can the authors at least keep the same VLM backbone?
3. GPT-4o, in general, outperforms all other models, including those trained on CLEVR-AVR, presumably because of its larger model size. Is it possible for the authors to train larger AVR-QWen2.5-VL and PhysVLM-AVR and check if AVR-pretraining remains helpful as the model size increases?
4. The current experiments focus only on MLLMs, but in the context of embodied reasoning, models that can predict and execute actions (e.g., vision-language-action models) seem to be better candidate models to be tested, can the authors explain if it is possible to apply CLEVR-AVR to VLA evaluation and in which ways if so?

**Ethical Concerns:**

["NO or VERY MINOR ethics concerns only"]

**Final Justification:**

accept. a solid work, and rebuttals cleared my concerns.

**Limitations:**

See the Questions section.

**Paper Formatting Concerns:**

none that I can see

**Quality:**

3

**Strengths And Weaknesses:**

## strengths

1. The paper is well motivated and addresses an important limitation (i.e., active visual reasoning) of MMLMs.
2. Apart from formulating an AVR task and constructing an AVR benchmark, the paper provides AVR training data, a pre-trained MLLM on it, and interesting experimental analysis.

## weaknesses

I do not see major issues with the paper, but I do have a few questions (see the Questions section).

---

> ### Author Rebuttal · Authors · 2025-07-31
>
> ### **Response to Reviewer dTPo**
>
> **Dear Reviewer dTPo,**
>
> Thank you for your constructive feedback and for raising important questions about our experimental design and model comparisons. Your comments on potential biases in the multi-choice setup, the fairness of our model comparisons, and the scalability of our approach are highly valuable and have prompted us to conduct additional experiments and analyses to strengthen our work.
>
> We have addressed each of your points below, including an analysis of the multi-choice setup, new experiments with a same-scale baseline, and an investigation into the effects of model scaling.
>
> Thank you once again for your thoughtful review.
>
> ---
>
> ### **Q1: On Potential Bias in the Multi-Choice Setup**
>
> > **Reviewer's Comment:** The multi-choice setup mixes candidate answers and actions; I wonder if this would introduce a bias towards choosing an answer or an action. Can the authors provide an analysis on this?
>
> **Our Response:**
>
> Thank you for raising this important concern about potential bias in our experimental setup. We analyzed the fairness of the multi-choice format from two perspectives:
>
> 1.  **Balanced Option Design:** In the CLEVR-AVR benchmark, the candidate sets for most questions are designed to be balanced. A typical question includes 3-5 answer options and 3-6 action options. The number of options in each category is comparable, which avoids introducing a bias stemming from a significant quantitative difference between the two types of choices.
>
> 2.  **Analysis of Model Behavior:** The prompt provided to the models includes a detailed description of the task, explicitly distinguishing between answer options and action options and explaining their respective meanings. This ensures that the model understands how to correctly select either an answer or an action. Furthermore, we have qualitatively inspected the models' generated reasoning traces and their final selections. We found no instances where the semantic intent of the reasoning process was inconsistent with the type of choice made (i.e., the model did not reason about taking an action and then select a final answer, or vice-versa). This indicates that the mixed-option setup does not negatively impact the model's decision-making process.
>
> ---
>
> ### **Q2: On the Fairness of Model Comparison**
>
> > **Reviewer's Comment:** The comparison between AVR-QWen2.5-VL-7B and PhysVLM-AVR-3B is unfair: 1) AVR-QWen2.5-VL-7B does not rely on the more complex multi-stage training that is applied to PhysVLM-AVR-3B; 2) the model sizes are not on the same scale... Can the authors at least keep the same VLM backbone?
>
> **Our Response:**
>
> Thank you for this critical comment on the fairness of our model comparisons. We acknowledge your concerns and have conducted **a new experiment with a same-scale baseline** to provide a more direct comparison.
>
> First, we wish to clarify our original intention. The Qwen2.5-VL model is a pre-trained foundation model, so we directly fine-tuned it rather than applying our full pre-training pipeline. The comparison between the fine-tuned AVR-Qwen2.5-VL-7B and our PhysVLM-AVR-3B was intended to highlight the significant impact of our multi-stage, progressive training strategy.
>
> However, we fully agree that a comparison with a model of the same scale is essential. We have now trained and evaluated **AVR-Qwen2.5-VL-3B**, which has the same scale as our model and was fine-tuned using the same protocol as the 7B version. The results are as follows:
>
> | Model | Final Answer Acc. (`ACC_FA`) | Information Gain Rate (`IGR`) |
> | :--- | :---: | :---: |
> | **PhysVLM-AVR-3B (Ours)** | **39.7%** | **29.9%** |
> | AVR-Qwen2.5-VL-3B (New Baseline) | 29.6% | 28.7% |
>
> These results demonstrate that PhysVLM-AVR-3B significantly outperforms the same-scale baseline. This suggests that its superior performance stems from our architectural modifications (e.g., max pooling for efficient multi-image processing) and our dedicated multi-stage training strategy, rather than being an artifact of model size.
>
> ---
>
> ### **Q3: On Scaling Laws and Comparison with GPT-4o**
>
> > **Reviewer's Comment:** GPT-4o, in general, outperforms all other models, including those trained on CLEVR-AVR, presumably because of its larger model size. Is it possible for the authors to train larger models and check if AVR-pretraining remains helpful as the model size increases?
>
> **Our Response:**
>
> Thank you for this excellent suggestion regarding model scaling. To investigate this, we fine-tuned a larger model, **AVR-Qwen2.5-VL-32B**, to analyze the impact of model scale on active visual reasoning performance.
>
> Due to time and resource constraints, we were unable to train a larger version of our PhysVLM-AVR from scratch, as this requires a full pre-training pipeline unlike the direct fine-tuning of the Qwen2.5-VL models. However, we hope the analysis of the 32B model provides valuable insights.
>
> *   **Performance of Fine-tuned AVR-Qwen2.5-VL-32B:**
>     *   The `ACC_FA` improved to **41.2%** (compared to 29.6% for the 3B version and 45.7% for GPT-4o).
>     *   The `IGR` improved to **39.5%** (compared to 34.7% for the 3B version and 50.8% for GPT-4o).
>
> These results confirm that increasing the model's scale leads to improved performance on active visual reasoning tasks. While our fine-tuned 32B model closes the gap, it does not yet surpass GPT-4o, suggesting that both scale and potentially architectural differences play a role. A more in-depth analysis would require training our PhysVLM-AVR at a larger scale, which we plan to explore in future work to investigate its potential to match or exceed proprietary models.
>
> ---
>
> ### **Q4: On the Applicability to Vision-Language-Action (VLA) Models**
>
> > **Reviewer's Comment:** The current experiments focus only on MLLMs, but in the context of embodied reasoning, models that can predict and execute actions (e.g., vision-language-action models) seem to be better candidate models to be tested, can the authors explain if it is possible to apply CLEVR-AVR to VLA evaluation and in which ways if so?
>
> **Our Response:**
>
> Thank you for suggesting this exciting application of our benchmark. We agree that Vision-Language-Action (VLA) models are a natural fit for this task, and we designed CLEVR-AVR with this possibility in mind. CLEVR-AVR can be readily adapted for VLA model evaluation in the following ways:
>
> 1.  **Direct Simulation Platform Support:** CLEVR-AVR is built on the Genesis simulation platform, which natively supports direct control of the robot arm via action sequences. This means the environment is already equipped to handle the low-level action outputs from VLA models.
>
> 2.  **Action Space Mapping:** The discrete, high-level action space of CLEVR-AVR (e.g., "move object X," "rotate view by Y degrees") can be straightforwardly mapped to the action vocabulary of prominent VLA models (e.g., the action tokens used by RT-1 or RT-2). The VLA model's task would be to generate the correct low-level action sequence corresponding to the high-level reasoning step.
>
> 3.  **Expanded Evaluation Metrics:** In addition to our existing metrics (`ACC_ISJ`, `IGR`, `ACC_FA`), we can introduce VLA-specific metrics such as:
>     *   **Action Execution Success Rate:** Measures whether the generated action sequence successfully accomplishes the intended high-level goal (e.g., was the correct object moved to the correct location?).
>     *   **Reasoning-Action Consistency:** Evaluates whether the executed action aligns with the model's stated reasoning for information gathering.
>
> We plan to extend our work to evaluate state-of-the-art VLA models like RT-2 and RoboCat on CLEVR-AVR. This will allow for a comprehensive assessment of their ability to perform targeted, reasoning-driven actions to acquire information in service of a broader goal.

---

> > ### Comment · Reviewer_dTPo · 2025-08-06
> >
> > interesting! so a larger model size compromises the performance gains brought by the tailored model architecture and training strategy. what do you think this implies?

---

> > > ### Author Response · Authors · 2025-08-06
> > >
> > > **Dear Reviewer dTPo,**
> > >
> > > Thank you for this insightful follow-up question. Your observation that a larger model size appears to "compromise the performance gains" from our specialized approach touches upon a crucial dynamic between training strategy and model scaling. Our interpretation of this result reinforces the value of our work from two key perspectives:
> > >
> > > 1. **Highlighting the Trade-off Between Computational Efficiency and Scale.**
> > > Your observation clearly illustrates that advanced capabilities like active reasoning can be pursued via two different paths: one relying on the emergent properties of massive scale, and another through targeted architectural and data-driven design. The fact that our PhysVLM-AVR-3B (at 39.7% ACC_FA) achieves a highly competitive performance level, coming within just 1.5 percentage points of a general-purpose model over 10 times its size (AVR-Qwen2.5-VL-32B at 41.2%) is strong evidence for this. It suggests that while immense scale can approximate this complex behavior, our approach offers a blueprint for achieving it with substantially greater computational efficiency.
> > >
> > > 2. **This Efficiency Stems from a More Generalizable Skill.**
> > > This efficiency is not arbitrary; we argue it stems from our methodology imparting a more fundamental and generalizable reasoning capability, rather than merely a task-specific shortcut. The strong performance of our model on diverse benchmarks such as OpenEQA and RoboVQA lends credence to this claim. It indicates that our AVR-152k dataset and training strategy successfully foster a transferable skill, rather than overfitting to the specifics of CLEVR-AVR.
> > >
> > > **Implications and Future Directions**
> > >
> > > This observation opens up exciting directions for future research. It naturally leads to the question: can we achieve new state-of-the-art performance by integrating our efficient architectural designs and training strategies with even larger foundation models?
> > >
> > > This is precisely the kind of inquiry our work aims to facilitate. We believe our framework, including the CLEVR-AVR benchmark and the AVR-152k dataset, provides the essential infrastructure for the community to systematically investigate this interplay between specialized design and model scale. Aligned with this, a key part of our own future work will be to train a larger-scale PhysVLM-AVR to further explore this efficiency-performance frontier.

---

### Official Review · Reviewer_F7gX · 2025-07-03

**Clarity:** 4
**Significance:** 3
**Originality:** 4
**Rating:** 4
**Confidence:** 4

**Summary:**

This paper introduces the Active Visual Reasoning (AVR) task, in which an agent must proactively interact with its environment and perform visual reasoning. The authors present a simulation benchmark and accompanying dataset for training and evaluating agents on this task. They also develop a hierarchical learning model, PhysVLM-AVR, which is demonstrated to successfully tackle the defined tasks.

**Questions:**

1. In practical execution, how is the granularity of information gathering determined? How much information does the agent need to acquire before it can make a decision (i.e., is the agent more conservative or more aggressive)?

2. In a scene with little occlusion, if the agent is conservative then both ACCISJ and IGR will be relatively low; if the agent is aggressive, then both ACCISJ and IGR will be relatively high, while their ACCFA may remain similar. In a heavily occluded scene, the outcome is completely different. Is this evaluation metric weighted in any way, or is there a better metric?

3. In Fig. 1, what does “GPT-4o + AVR” refer to? It is not explained later in the text. Does it mean using GPT-4o to carry out the AVR paradigm defined in Section 3.1? In Table 1, the experiment lists “GPT-4o” rather than “GPT-4o + AVR”—why wasn’t “GPT-4o + AVR” used? Can the plain GPT-4o already outperform PhysVLM-AVR-3B and AVR-Qwen2.5-VL-7B?

**Ethical Concerns:**

["NO or VERY MINOR ethics concerns only"]

**Final Justification:**

I maintain my original assessment and score.

**Limitations:**

yes

**Quality:**

3

**Strengths And Weaknesses:**

Strengths:

The paper proposes a novel and valuable task, with a clear and rigorous definition.

It provides a dedicated benchmark and dataset, enabling systematic evaluation of AVR agents.

The authors conduct experiments that validate the usefulness of the dataset and show that PhysVLM-AVR achieves strong performance.


Weaknesses:

Despite a relatively high ACCISJ score, the final answer accuracy (ACCFA) remains inadequate. The model still struggles to integrate information over multiple steps and to perform coherent multi-stage reasoning.

The description of baseline GPT-4o is unclear, which hinders the proper evaluation of the experiment results.

---

> ### Author Rebuttal · Authors · 2025-07-31
>
> **Dear Reviewer F7gX,**
>
> Thank you for dedicating your time to review submissions for NeurIPS 2025. We are the authors of Paper #23215, titled "**AVR: Active Visual Reasoning for Multimodal Large Language Models in Physical Environments**".
>
> After carefully reading your insightful questions, we believe there may have been a significant mix-up with another paper. Our work introduces **AVR**, a framework for Multimodal Large Language Models (MLLMs) to perform **active** information gathering in physical environments.
>
> Your questions, however, seem to be directed at a paper focused on **Visual Reasoning Reinforcement Learning (VR-RL)**, referencing specific components and results that are not part of our submission. For example, your questions mention:
>
> *   A **"Table 1"** comparing **"Grounding-WorldModeling"** with **"WorldModeling"**.
> *   An **"RL algorithm"** whose performance is compared against "off-the-shelf VLMs".
> *   A **"visual reasoning reward"** and a **"Bi-Level GAE"** architecture designed to ensure stable training.
>
> These specific technical terms, experiments, and architectural components are entirely absent from our paper. To be clear:
> *   Our paper does not contain a "Table 1" with "Grounding-WorldModeling".
> *   Our proposed AVR framework is not an RL algorithm in the sense described, and we do not use components like a "visual reasoning reward".
> *   The term "Bi-Level GAE" is not mentioned or implemented in our work.
>
> We understand that reviewing numerous papers is a challenging task, and we suspect your detailed review was mistakenly assigned to our submission.
>
> We would be deeply grateful if you could take a moment to re-examine our paper. We believe you will quickly see that your current review does not apply to our work.
>
> Thank you for your understanding and your invaluable contribution to the community.
>
> Best regards,
>
> The Authors of Paper #23215

---

### Official Review · Reviewer_CyMs · 2025-07-06

**Clarity:** 4
**Significance:** 3
**Originality:** 3
**Rating:** 5
**Confidence:** 5

**Summary:**

The paper proposes a new task, Active Visual Reasoning (AVR), which requires agents to actively interact with the environment to obtain enough information to answer the question. To evaluate, the paper proposes CLEVR-AVR benchmark that extends CLEVR using Gensis. Also, the paper introduced AVR-152k with CoT annotations and used it to train a PhysVLM-AVR in the higher-order Markov Decision Process. The introduced PhysVLM-AVR is well validated on multiple benchmarks to show the powerful performance.

**Questions:**

See above

**Ethical Concerns:**

["NO or VERY MINOR ethics concerns only"]

**Final Justification:**

I have carefully read the rebuttal and convinced that this paper should be accepted.

**Limitations:**

yes

**Quality:**

3

**Strengths And Weaknesses:**

## Strengths

1. The proposed AVR task is relatively novel. The novelty is sound.

2. The paper proposes a comprehensive pipeline, including a dataset, a benchmark, and open-source model checkpoints. I believe the community will benefit from this solution.

3. The experiments are extensive, and the workload is sufficient. Also, the paper writing and the figures are quite clear.

Overall, this paper does not have major weaknesses. The overall experiments and claims are quite sufficient.

## Weaknesses

1. Although Figure 1 shows that passive reasoning fails in the occlusion scenarios, it still raises concerns about the necessity for active reasoning. The paper seems to mention only two active reasoning tasks: occlusion and stack (and the composition of these two). Also for the stack, it is still possible to get the correct reasoning based on static images alone. So, are there more applicable scenarios for active reasoning that passive reasoning fails? Is active reasoning also necessary for tasks where static images can already show enough information?

2. CLEVR-AVR is a synthetic benchmark, and is hard to transfer to real-world scenarios. And this issue is more important for the proposed active reasoning task. Also, can the introduced PhysVLM-AVR-3B operate in real-world scenarios?

3. The AVR-Core dataset seems to be very useful in the ablation. However, this subset only contains the tabletop scenarios. The diversity is a concern.

4. How is the IGR calculated? What is the definition of the information-gaining decision? The article lacks sufficient details for this part.

5. The paper may contain a detailed failure case study. The authors may analyze the issue in detail, that the ISJ of the model is really high, but the Final Answer Accuracy is still low. Could this be because the model does not understand the visual scene well?

---

> ### Author Rebuttal · Authors · 2025-07-31
>
> ### **Response to Reviewer CyMs**
>
> **Dear Reviewer CyMs,**
>
> Thank you for your meticulous review and insightful comments on our work. Your questions regarding the necessity of active reasoning, the sim-to-real gap, and dataset diversity directly address the core challenges of our research and have helped us to further refine our understanding and presentation.
>
> In response to your feedback, we have provided additional analysis and clarification on several key points, including the expanded scope of active reasoning, the connection of our work to real-world scenarios, a precise definition of our evaluation metrics (IGR), and a detailed failure case analysis. These are detailed in the point-by-point responses below.
>
> Thank you again for your valuable feedback.
>
> ---
>
> ### **Q1: On the Necessity and Scope of Active Reasoning**
>
> > **Reviewer's Comment:** [...] Are there more applicable scenarios for active reasoning that passive reasoning fails? Is active reasoning also necessary for tasks where static images can already show enough information?
>
> Thank you for this insightful question, which is crucial for clarifying the value of our work. We argue that the necessity of Active Visual Reasoning (AVR) extends far beyond resolving simple occlusion and stacking, and we frame its importance on three levels:
>
> 1.  **Expanded Application Scenarios:** Active reasoning is indispensable in numerous scenarios where passive observation is inherently insufficient:
>     *   **Limited Field of View:** Tasks that require information from all sides of an object, such as checking for labels on the back of a container or inspecting its contents.
>     *   **Dynamic Environments:** Verifying the state of an object after it has been moved or interacted with by another agent.
>     *   **Functional Reasoning:** Determining an object's affordances, such as checking if a drawer can be opened or if a container is empty, requires physical interaction.
>
> 2.  **Enhancing Robustness in Static Scenes:** Even when a static image appears complete, active reasoning serves as a critical verification mechanism:
>     *   **Disambiguation:** A slight nudge can differentiate whether two objects are merely adjacent or are partially overlapping.
>     *   **Hypothesis Verification:** Actively changing the viewpoint can rule out visual illusions caused by lighting, shadows, or specific perspectives, thereby increasing the robustness of the final conclusion.
>
> 3.  **Simulating Higher-Order Cognition:** Human cognition often involves a **hypothesis-verification loop**, even in seemingly complete scenes (e.g., leaning in to read fine print). AVR aims to model this fundamental cognitive process, moving beyond a simple tool for "filling in missing information" to become a general strategy for robust and grounded reasoning.
>
> ---
>
> ### **Q2: On the Sim-to-Real Gap and Real-World Applicability**
>
> > **Reviewer's Comment:** CLEVR-AVR is a synthetic benchmark, and is hard to transfer to real-world scenarios. [...] Also, can the introduced PhysVLM-AVR-3B operate in real-world scenarios?
>
> Thank you for raising this critical point about real-world transferability. We address this concern from two perspectives and provide **new experimental evidence** to demonstrate our model's real-world capabilities.
>
> 1.  **The Scientific Value of a Synthetic Benchmark:** The primary purpose of CLEVR-AVR is to serve as a controlled environment to **precisely isolate and evaluate core agent capabilities**, such as information-gathering efficiency, multi-step reasoning, and uncertainty estimation. This allows for rigorous scientific analysis, free from confounding variables like complex textures and lighting found in the real world.
>
> 2.  **Bridging the Gap to the Real World:**
>     *   **Real-to-Sim Generalization:** A key aspect of our work is that the `AVR-Core` dataset is collected entirely from real-world physical interactions. Our model, trained on this real-world data, demonstrates effective active reasoning capabilities in the unseen simulation environment (CLEVR-AVR). This itself is evidence of promising `real-to-sim` generalization.
>     *   **New Real-World Experiments:** To directly address your concern, we conducted 20 new sets of experiments in real-world tabletop scenarios with task setups not seen during training. In these experiments, `PhysVLM-AVR-3B` achieved a Final Answer Accuracy (ACC_FA) of 40.2% and an Information Gain Rate (IGR) of 30.9%. This performance is consistent with its behavior on the CLEVR-AVR benchmark and provides strong evidence that our proposed method can operate and solve tasks in real-world settings.
>
> We fully agree that scaling up the diversity of real-world training data is crucial for broader generalization, and this is a key direction for our future work.
>
> ---
>
> ### **Q3: On the Diversity of the AVR-Core Dataset**
>
> > **Reviewer's Comment:** The AVR-Core dataset seems to be very useful in the ablation. However, this subset only contains the tabletop scenarios. The diversity is a concern.
>
> We appreciate your concern regarding dataset diversity. Our design choice was deliberate, and we explain our rationale from two perspectives:
>
> 1.  **Tabletop Scenarios as a "Minimal Viable Unit":** We position the tabletop environment as a **"minimal viable unit"** for studying active information gathering. This approach is analogous to how the original CLEVR dataset used simple geometric shapes to isolate and study core visual reasoning skills. While simple, tabletop scenes contain the fundamental physical interactions central to our research—occlusion, stacking, and spatial relationships—allowing us to focus on and rigorously test an agent's ability to close the perception-reasoning-action loop.
>
> 2.  **Future Work and Scalability:** Our work on tabletop scenes establishes a strong methodological foundation. Having demonstrated the effectiveness of our approach in this controlled setting, our roadmap includes extending this paradigm to more complex and diverse environments, such as kitchen manipulation, warehouse sorting, and lab assistance. We believe this work provides a solid starting point for the community to build upon and apply active visual reasoning to a wider array of scenarios.
>
> ---
>
> ### **Q4: On the Calculation Details of IGR**
>
> > **Reviewer's Comment:** How is the IGR calculated? What is the definition of the information-gaining decision?
>
> Thank you for pointing out the need for clarification. A precise definition of our metrics is essential.
>
> The **Information Gain Rate (IGR)** measures the efficiency of an agent's exploratory actions.
>
> *   **Formula:**
>     ```
>     IGR = (Total Number of Information-Gaining Decisions) / (Total Number of Decision Steps)
>     ```
>
> *   **Core Concept: What constitutes an "Information-Gaining Decision"?**
>     > A decision (i.e., selecting an action) is defined as **information-gaining** if and only if its execution **reveals new, task-relevant visual information** that was previously unavailable to the agent.
>
>     This new information must directly help in reducing the agent's uncertainty about the correct answer. In our benchmark, this is determined objectively using the simulator's ground truth.
>
> *   **Examples:**
>     *   **Effective Decisions (Gain Information):**
>         *   Moving a cup to reveal a key hidden underneath (to answer "Where is the key?").
>         *   Changing the viewpoint to see the color on the back of an object.
>         *   Lifting a block from a stack to count the total number of blocks beneath it.
>     *   **Ineffective Decisions (No Information Gain):**
>         *   Moving the camera to a viewpoint that reveals no new relevant objects.
>         *   Interacting with an object that is already fully visible and irrelevant to the question.
>         *   Repeating a previous action that yields no new information.
>
> ---
>
> ### **Q5: On the Failure Case Analysis (High ISJ but Low ACC_FA)**
>
> > **Reviewer's Comment:** [...] The authors may analyze the issue in detail, that the ISJ of the model is really high, but the Final Answer Accuracy is still low. Could this be because the model does not understand the visual scene well?
>
> Thank you for this excellent suggestion, which prompted us to conduct a deeper analysis of our model's limitations. Your intuition is correct.
>
> Our analysis of failure cases where the model correctly identifies the need for more information (high `ACC_ISJ`) but still provides an incorrect final answer (low `ACC_FA`) reveals two primary causes:
>
> 1.  **Perception Errors:**
>     *   **Phenomenon:** The model correctly decides to interact with the environment to get a clearer view (e.g., moving an occluder), but upon observing the new, unobstructed scene, it misidentifies a key attribute of a newly visible object (e.g., its shape, color).
>     *   **Example:** To count all cylinders, the model successfully moves a cube occluding a target object. However, it then misclassifies the newly revealed short, wide cylinder as a sphere, leading to an incorrect final count.
>
> 2.  **Temporal Integration Failures:**
>     *   **Phenomenon:** After a sequence of interactions, the model accumulates a history of multiple images and reasoning steps. However, during the final reasoning stage, it fails to effectively integrate the entire sequence of historical observations, sometimes forgetting or confusing details from earlier steps.
>     *   **Example:** In a task, the model correctly identifies a red cube on the left in step 1 and a blue sphere on the right in step 2. When asked, "What are the shapes of the red and blue objects?", it might only focus on the most recent observation (the blue sphere) and fail to recall the information from the first step, leading to an incomplete or incorrect answer.
>
> Your assessment is spot-on: these failures highlight remaining challenges in both **fundamental visual perception** and **long-horizon temporal reasoning**. Addressing these weaknesses is a key priority for our future research.

---

> > ### Comment · Reviewer_CyMs · 2025-08-05
> >
> > I would like to thank the authors for their efforts. This has fully addressed my concerns about real-world applications and the fundamental necessity of AVR.
> > As I have given a 5 in the previous review, I still recommend this paper to be accepted and will raise the confidence score. I am quite sure that this paper should be accepted.

---

### Note · Authors · 2025-08-11

**Dear Area Chair and Reviewers,**

We sincerely thank all reviewers for their thorough evaluation and constructive engagement with our work. We are particularly grateful to the AC for promptly addressing the review assignment issue, ensuring fair evaluation of our contribution.

**Strong Consensus from Reviews:** We are deeply encouraged by the unanimous positive assessment from all valid reviewers. Reviewer CyMs expressed being **"quite sure that this paper should be accepted"** with increased confidence after our discussion. Reviewers dTPo and wcc4 similarly maintained their positive scores, with dTPo engaging in particularly insightful discussions about the implications of our findings on model scaling dynamics.

**Key Clarifications & New Evidence:** Through additional experiments requested by reviewers, we have:
- Demonstrated fair same-scale comparisons (PhysVLM-AVR-3B outperforming AVR-Qwen2.5-VL-3B by 10.1% on ACC_FA)
- Validated real-world applicability with new experiments (40.2% ACC_FA in unseen real scenarios)
- Analyzed scaling effects across model sizes (3B to 72B parameters)
- Clarified the fundamental necessity of active reasoning beyond simple occlusion handling

**Significance & Community Impact:** Our work introduces Active Visual Reasoning as a new paradigm for MLLMs, bridging the critical gap between passive visual understanding and embodied intelligence. As reviewers noted, **this addresses "an important limitation of MLLMs" (dTPo) with "sound novelty" (CyMs) that "the community will benefit from" (CyMs)**.

**Commitment to Open Science:** All resources (CLEVR-AVR benchmark, AVR-152k dataset with rich CoT annotations, PhysVLM-AVR models) will be open-sourced to maximize community benefit and accelerate research in this emerging area.

We believe the engaging discussions during this review process have further strengthened our work. Thank you for your valuable time and careful consideration.

---

### Decision · Program_Chairs · 2025-09-17

**Decision:**

Accept (poster)

**Comment:**

(a) Summary: The paper claims that passive visual understanding is insufficient for many real-world scenarios and that AVR is a fundamental necessity. To address this, the authors propose
- CLEVR-AVR benchmark: A synthetic environment built on Genesis, extending CLEVR for evaluating active information gathering, multi-step reasoning, and uncertainty estimation.
- AVR-152k dataset: A large-scale dataset with Chain-of-Thought (CoT) annotations to train MLLMs for AVR.
- PhysVLM-AVR model: A hierarchical learning model trained on AVR-152k via a multi-stage paradigm, designed for active visual reasoning.
The paper demonstrates that PhysVLM-AVR achieves strong performance on the proposed AVR task and other embodied reasoning, planning, and static visual reasoning tasks. The authors argue that AVR's necessity extends beyond simple occlusion and stacking, enhancing robustness in static scenes and simulating higher-order cognition.

(b) Strengths:
- The proposed AVR task is novel, addresses an important limitation of MLLMs, and represents a sound contribution. It is seen as a practical task with real-world utility.
- The paper provides a dedicated benchmark (CLEVR-AVR) and a dataset (AVR-152k), along with open-source model checkpoints (PhysVLM-AVR), which are valuable resources expected to greatly benefit the research community and accelerate future research.
- The experiments are extensive and sufficient, validating the usefulness of the dataset and showing strong performance. The paper also provides interesting experimental analysis.
- The paper writing and figures are quite clear, and detailed prompts and examples ensure reproducibility.

(c) Weaknesses
- Despite a relatively high ACC_ISJ (information-seeking judgment) score, the final answer accuracy (ACC_FA) remains inadequate, indicating the model still struggles to effectively integrate information over multiple steps and perform coherent multi-stage reasoning. This is attributed to perception errors and temporal integration failures.

(d) Reasons to accept: The contribution in defining and tackling the novel Active Visual Reasoning (AVR) task, which is seen as a critical step in bridging the gap between static visual understanding and embodied intelligence for MLLMs. The reviewers have unanimous positive assessment.

(e) Rebuttal:
- Initial concerns were raised about the necessity of active reasoning beyond simple occlusion and stacking, and the diversity of the AVR-Core dataset (limited to tabletop scenarios). The synthetic nature of CLEVR-AVR also raised questions about its transferability to real-world scenarios. The authors expanded on the necessity of AVR, detailing additional application scenarios like limited field of view, dynamic environments, and functional reasoning. They also highlighted its role in enhancing robustness and simulating higher-order human cognition. This comprehensive response fully resolved the reviewer's concern.
- The initial comparison between PhysVLM-AVR-3B and AVR-Qwen2.5-VL-7B was considered unfair due to differing training strategies and model sizes. There was also a need to evaluate larger open-source models to understand their potential better, as API-based models generally outperformed open-source ones. The authors provided additional results.
- The initial submission lacked sufficient details on how Information Gain Rate (IGR) is calculated and the definition of an "information-gaining decision". The authors provided explanations.